# Separating the role of direct radiative heating and photolysis in modulating the atmospheric response to the amplitude of the 11-year solar cycle forcing

Ewa M. Bednarz[1,a], Amanda C. Maycock[1,2,b], Peter Braesicke[1,2,c], Paul J. Telford[1,2], N. Luke Abraham[1,2] and John A. Pyle[1,2]

[1] Department of Chemistry, University of Cambridge, Cambridge, UK
[2] National Centre for Atmospheric Science - Climate, UK
[a] now at: Lancaster Environment Centre, Lancaster University, Lancaster, UK
[b] now at: School of Earth and Environment, University of Leeds, Leeds, UK
[c] now at: Karlsruhe Institute of Technology, Institute for Meteorology and Climate Research, Karlsruhe, Germany

Correspondence to: Ewa Bednarz (e.bednarz@lancaster.ac.uk)

## Abstract

The atmospheric response to the 11-year solar cycle is separated into the contributions from changes in direct radiative heating and photolysis rates using specially designed sensitivity simulations with the UM-UKCA chemistry-climate model. We perform a number of idealised timeslice experiments under perpetual solar maximum (SMAX) and minimum conditions (SMIN), and find that contributions from changes in direct heating and photolysis rates are both important for determining the stratospheric shortwave heating, temperature and ozone responses to the amplitude of the 11-year solar cycle. The combined effects of the processes are found to be largely additive in the tropics but non-additive in the Southern Hemisphere (SH) high latitudes during the dynamically active season. Our results indicate that, in contrast to the original mechanism proposed in the literature, the solar-induced changes in the horizontal shortwave heating rate gradients not only in autumn/early winter, but throughout the dynamically active season are important for modulating the dynamical response to changes in solar forcing. In spring, these gradients are strongly influenced by the shortwave heating anomalies at higher southern latitudes, which are closely linked to the concurrent changes in ozone. In addition, our simulations indicate differences in the winter SH dynamical responses between the experiments. We suggest a couple of potential drivers of the simulated differences, i.e. the role of enhanced zonally-asymmetric ozone heating brought about by the increased solar-induced ozone levels under SMAX and/or sensitivity of the polar dynamical response to the altitude of the anomalous radiative tendencies. All in all, our results suggest that solar-induced changes in ozone, both in the tropics/mid-latitudes and the polar regions, are important for modulating the SH dynamical response to the 11-year solar cycle. In addition, the markedly non-additive character of the SH polar vortex response simulated in austral spring highlights the need for consistent model implementation of the solar cycle forcing in both the radiative heating and photolysis schemes.

# 1. Introduction

It is now well understood that changes in the incoming ultraviolet (UV) radiation associated with the 11-year solar cycle influence temperatures and ozone concentrations across much of the stratosphere (e.g.: Penner and Chang, 1978; Brasseur and Simon, 1981; Haigh, 1994; Randel et al., 2009; Ramaswamy et al., 2001; Keckhut et al., 2005; Soukharev and Hood, 2006; Mitchell et al., 2015b; Maycock et al., 2016). In addition to being a major driver of decadal variability within the stratosphere, these effects can initiate a dynamical response that propagates down into the troposphere (e.g.: Kuroda and Kodera, 2002; Kodera and Kuroda, 2002), thereby affecting surface climate variability (e.g. Thieblémont et al., 2015). The incoming UV radiation (increased for enhanced solar cycle activity) is absorbed in the middle atmosphere by oxygen and ozone molecules, the photolysis of which (Eq. 1, Eq. 3) leads to formation of ozone, predominantly in the stratosphere, and shortwave heating (Eq. 2):

$$O_2 + h\nu \ (\lambda < 242 \text{ nm}) \rightarrow O + O \tag{1}$$

$$O + O_2 + M \rightarrow O_3 + M \quad (M = N_2, O_2, \ldots) \tag{2}$$

$$O_3 + h\nu \ (\lambda < 1180 \text{ nm}) \rightarrow O + O_2 \tag{3}$$

Clearly, the heating of the stratospheric air parcels through direct absorption of solar radiation by ozone and the photochemical production of ozone are closely coupled. However, this is not necessarily the case in atmospheric models. In chemistry-climate models (CCMs), shortwave heating from ozone is usually handled by the radiation scheme, a crucial physical component of any climate model. A photochemistry module in turn solves the chemical reactions that lead to ozone production. The accuracy of individual schemes, as well as the method for implementing the solar cycle forcing, can vary substantially between models (e.g. SPARC, 2010; Sukhodolov et al., 2016). Such differences are likely to affect the simulated responses to the solar cycle forcing across different CCMs. Furthermore, not all climate models include an interactive chemistry module and, therefore, are capable of including a feedback from ozone that is consistent with the imposed spectral solar irradiance (SSI) changes and the resulting adjustments of temperature and transport. In general, there has been a wide spread of modelled atmospheric responses to the 11-year solar cycle forcing reported in the literature (e.g.: Austin et al., 2008; SPARC, 2010; Mitchell et al. 2015a; Hood et al., 2015). A number of these multi-model studies have attempted to attribute the spread of modelled atmospheric responses to the solar cycle forcing to the details of specific aspects of model design (e.g. the resolution of the radiation scheme; height of model top… etc.); such a task is, however, inherently difficult owing to the wide diversity in model design.

In the spirit of understanding the contributions of modelled radiation and photolysis processes to the simulated 11-year solar cycle response, this paper examines the responses to the amplitude of the 11-year solar cycle with the forcing included separately in either the radiation or photolysis scheme. While some studies reported results of similar calculations made with fixed dynamical heating (FDH) models (e.g.: Shibata and Kodera, 2005; Gray et al., 2009) or, for only the annual mean

using a CCM (e.g. Swartz et al., 2012), separating the impacts of these processes on the 11-year solar cycle response at seasonal timescales has not, to our knowledge, received much attention in the literature. Clearly such decomposition is, by definition, an idealised study owing to the strong physical coupling between the radiative and photochemical processes in the atmosphere. However, this is a valuable exercise as it helps to elucidate the factors that can affect the modelled response to

the 11-year solar cycle forcing, and thus whether these may contribute to the divergent multi-model results described above.

We focus here on the direct responses to the solar cycle forcing in the tropics (yearly mean), as well as on the corresponding circulation responses in the Southern Hemisphere (SH) during winter/spring. It is now well established that the SH high latitude stratosphere experiences on average lower wave activity than the Northern Hemisphere (NH). This makes the SH

polar vortex stronger, less variable on interannual timescales and closer to the thermodynamical equilibrium than its NH counterpart, thereby enhancing the detection of the solar-induced anomalies in the region. In addition, while the solar-induced dynamical response in the NH, including its underpinning mechanisms, has received considerable attention in the literature (e.g.: Yukimoto and Kodera, 2007; Ineson et al., 2011; Scaife et al., 2013; Andrews et al., 2015; Gray et al., 2016), the corresponding SH dynamical response and the mechanisms driving it are not as extensively examined (e.g. Haigh and

Roscoe, 2006; Kuroda and Shibata, 2006;  Kuroda et al., 2007; Petrick et al., 2012; Kuroda and Deuschi, 2016).

Section 2 discusses the model and experiments used. Section 3 introduces the yearly mean temperature responses to the amplitude of the 11-year solar cycle with the forcing included exclusively in either the radiation or photolysis scheme, as compared to the control case that includes them both, and points out the key regions discussed in this paper. Section 4

discusses the tropical yearly mean responses in the simulations performed, and Section 5 the corresponding SH dynamical responses in winter and spring. This is followed by a consideration of a potential explanatory mechanism for the different effects of the solar cycle forcing in the photolysis and radiation schemes (Section 6) and the discussion of the results (Section 7). The paper is summarised in Section 8.

## 2. The experiments

We use the United Kingdom Chemistry and Aerosol Model coupled to version 7.3 of the Met Office Unified Model (UM-UKCA) in the atmosphere-only HadGEM3-A r2.0 configuration (Hewitt et al., 2011). The chemistry scheme used is the extended Chemistry of the Stratosphere Scheme (CheS+), as described in Bednarz et al. (2016). Unlike in Bednarz et al. (2016), however, the model version used here does not include the coupling of stratospheric aerosols with the radiation and photolysis schemes.


The implementation of the 11-year solar cycle forcing in the radiation and photolysis schemes is identical to that described in Bednarz et al. (2019). The yearly mean total solar irradiance (TSI) data used are those recommended for the CMIP5

(Coupled Model Intercomparison Project 5) simulations (Fröhlich and Lean, 1998; Lean, 2000; Wang et al., 2005; Lean, 2009), processed to force the mean of the 1700-2004 period to be 1365 $Wm^{-2}$ (Jones et al., 2011). A fit to spectral data from Lean (1995) is used by the radiation scheme to account for the change of partitioning of solar radiation into wavelength bins. In the Fast-JX photolysis scheme used here (Telford et al., 2013), the change in partitioning of solar irradiance into

wavelength bins is accounted for by scaling the photolysis bins according to the difference in the yearly mean CMIP5 SSI data for the years 1981 and 1986 (solar maximum, SMAX, and solar minimum, SMIN, respectively), and the long-term evolution of the processed TSI. A more detailed description of the implementation of the 11-year solar cycle variability in UM-UKCA, including an evaluation of the atmospheric response to the 11-year solar cycle forcing, can be found in Bednarz et al. (2019). Note that since the model uses prescribed SSTs, the full tropospheric response to the imposed change in solar

forcing will not be captured, as tropospheric temperatures are strongly constrained by the imposed SSTs.

Long timeseries are needed in order to confidently diagnose the atmospheric response to the 11-year solar cycle forcing. Therefore, in order to increase the signal-to-noise ratio while minimising the computational requirements of long transient integrations, a number of perpetual-year "timeslice" integrations are performed under either perpetual SMAX or SMIN

conditions. These are represented by the annual mean solar forcing conditions for the years 1981 and 1986, respectively ($\Delta$TSI = 1.06 $Wm^{-2}$). All other forcings are climatological and identical in all runs. These include the 1977-1987 mean of the SSTs and sea-ice (Rayner et al., 2003), and of surface and aircraft emissions of CO, HCHO (both surface-only) and $NO_x$ following the CCMVal2 (Chemistry-Climate Model Validation 2) specifications (Morgenstern et al., 2010). The levels of greenhouse gases and ozone-depleting substances for the year 1982 are used according to the SRES A1B scenario (IPCC,

2000) and WMO (2011), respectively. Ozone (as well as $N_2O$, $CH_4$, $CCl_3F$, $CCl_2F_2$, $C_2Cl_3F_3$ and $CHClF_2$) in all runs is treated interactively, i.e. the chemical ozone field, transported by the circulation, is also used by the radiation scheme.

We present the results from six 50-year-long (+10 years spin-up) integrations combined into three SMAX/SMIN pairs. The first pair, INTERO3$_{SMAX/SMIN}$, represents the control case with the 11-year solar cycle forcing implemented consistently in

both the radiative heating and photolysis schemes. In the second pair, RAD-ONLY$_{SMAX/SMIN}$, the solar cycle forcing is implemented exclusively in the radiative heating scheme. In the photolysis scheme, no solar cycle modulation of the spectral distribution is used in either SMAX and SMIN, but note that the indirect impact on ozone through changes in atmospheric temperatures and transport will be captured. The third pair, PHOT-ONLY$_{SMAX/SMIN}$, is analogous to RAD-ONLY$_{SMAX/SMIN}$, but the solar cycle forcing is included exclusively in the photolysis scheme while constant TSI and SSI are used in the

radiation scheme. Importantly, as noted above, the perturbed ozone field from the photochemistry is passed to the radiation scheme and will therefore couple back onto climate. We analyse the resulting differences between the simulated SMAX and SMIN responses for each pair and, for brevity, henceforth we refer to them without any subscripts as INTERO3, RAD-ONLY and PHOT-ONLY. The experimental set-up is summarised in Table 1.

| Experiment | Length (years) | Solar cycle phase | Solar forcing in radiation | Solar forcing in photolysis |
|---|---|---|---|---|
| INTERO3$_{SMAX}$ | 50 | MAX | Yes | Yes |
| INTERO3$_{SMIN}$ | 50 | MIN | Yes | Yes |
| PHOT-ONLY$_{SMAX}$ | 50 | MAX | No | Yes |
| PHOT-ONLY$_{SMIN}$ | 50 | MIN | No | Yes |
| RAD-ONLY$_{SMAX}$ | 50 | MAX | Yes | No |
| RAD-ONLY$_{SMIN}$ | 50 | MIN | Yes | No |

**Table 1. Summary of the sensitivity timeslice experiments performed.**

## 3. The yearly mean temperature response

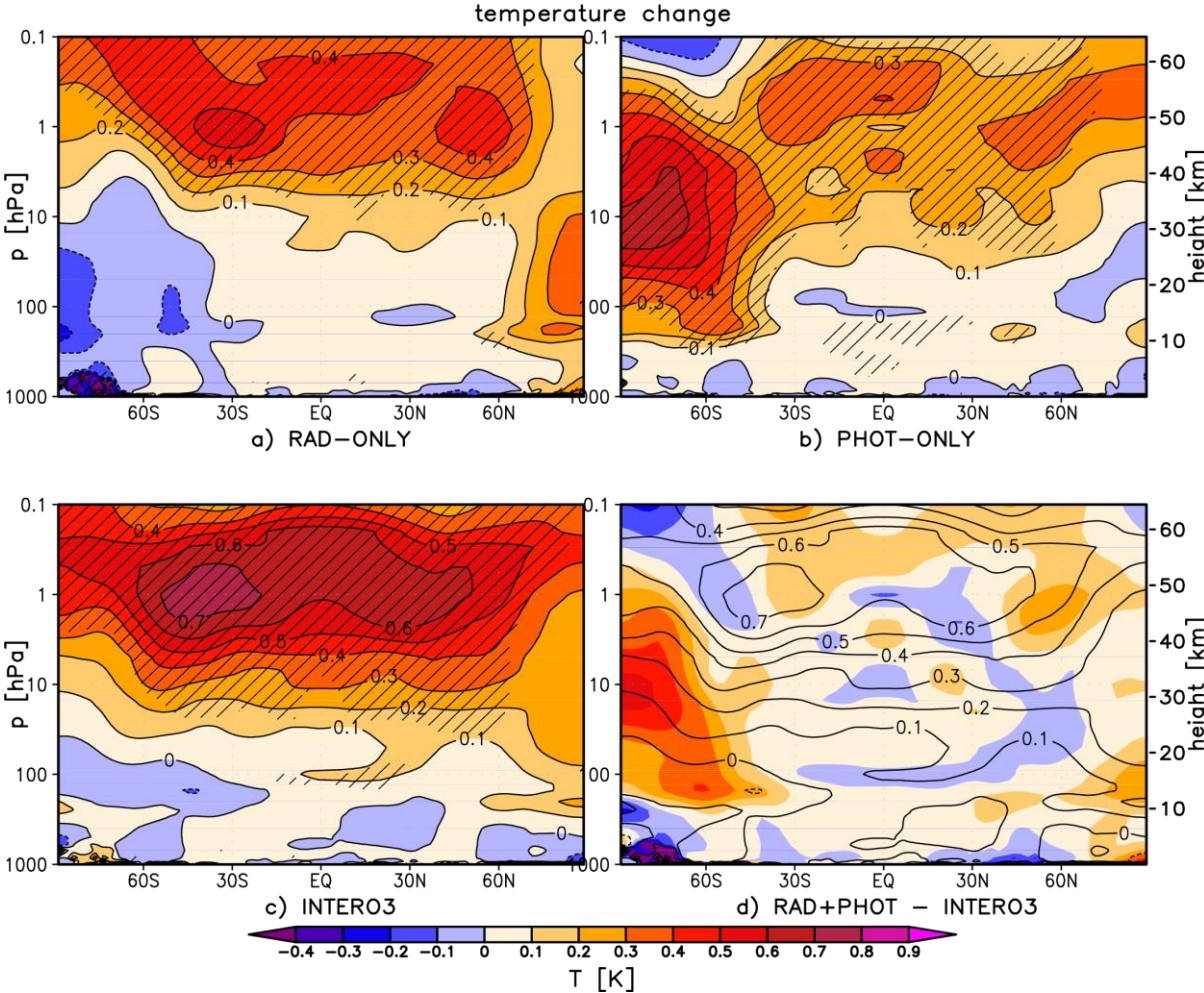

**Figure 1. Yearly mean zonal mean temperature change [K] between SMAX and SMIN for (a) RAD-ONLY, (b) PHOT-ONLY and (c) INTERO3. Hatching in (a-c) shows regions where the response is statistically significant at the 95% level (calculated using a two-tailed Student's t-test). Shown also (d) is the difference (shading) between the sum of the single forcing responses and INTERO3 (contours, as in (c)).**

Figure 1 shows the simulated yearly mean SMAX-SMIN temperature responses in the single forcing experiment pairs (RAD-ONLY and PHOT-ONLY, a-b) and in the control pair with both forcings included (INTERO3, c). In RAD-ONLY, the temperature response maximises near the tropical and mid-latitude stratopause at ~0.4-0.5 K. In PHOT-ONLY, the response simulated in this region is somewhat smaller (up to ~0.3-0.4 K); its magnitude also decreases less rapidly with decreasing altitude. With the exception of a small overestimation in the tropical lower mesosphere, the response obtained by combining the single forcing responses in the tropics agrees with the response in the control pair (up to ~0.6-0.7 K, d).

Importantly, the individual responses to direct radiative heating and photolysis cannot be linearly combined to capture the total response in the high latitudes, in particular in the SH. The stratospheric temperature increase in INTERO3 decreases slowly at latitudes poleward of 60° in both hemispheres (Fig. 1c). In contrast, PHOT-ONLY shows a distinct yearly mean warming of the SH polar stratosphere (up to ~0.6 K). The magnitude of this polar temperature response exceeds that found near the tropical stratopause. In comparison, the yearly mean temperature in RAD-ONLY does not change substantially throughout most of the Antarctic stratosphere. The sum of the yearly mean RAD-ONLY and PHOT-ONLY responses ('RAD+PHOT') over the Antarctic shows up to ~0.5 K difference to the INTERO3 response, Fig. 1d. This is large enough to exceed the ±2 standard error confidence interval around the INTERO3 response (not shown), although we note that the difference between RAD+PHOT and INTERO3 responses is not significant in a strict statistical sense when the confidence intervals around both RAD+PHOT[1] and INTERO3 are considered.

We therefore concentrate in this paper on two regions: firstly the tropics, where the stratospheric responses appear mostly linearly additive, and secondly the SH high latitudes, where they do not.

## 4. The tropical yearly mean response

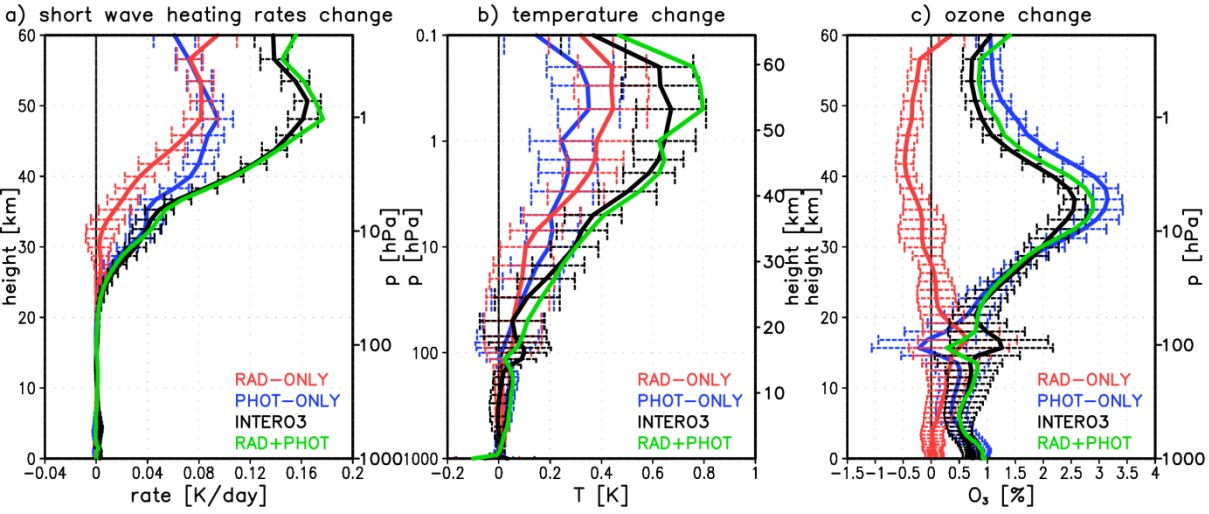

**Figure 2. Yearly mean tropical average (25°N-25°S) change in (a) the shortwave heating rates [K day⁻¹], (b) temperature [K], and (c) ozone [%] between SMAX and SMIN for RAD-ONLY (red), PHOT-ONLY (blue) and INTERO3 (black), together with the associated confidence intervals (±2 standard errors). The green line indicates the sum of the RAD-ONLY and PHOT-ONLY responses.**

---

[1] where the standard errors in PHOT-ONLY and RAD-ONLY are added in quadrature

## 4.1. Shortwave heating rates

Figure 2a shows the yearly mean tropical mean (25°S-25°N) SMAX-SMIN differences in shortwave heating rates (SWHRs) in the three pairs of experiments. In RAD-ONLY, the SWHRs increase directly due to the increased solar radiation and the resulting enhanced absorption by ozone. In PHOT-ONLY, even though the prescribed SSI does not change in the radiative
scheme calculations, the increased levels of ozone (Section 4.3; Fig. 2c) enhance the SWHR, as described by Haigh (1994).

The maximum amplitudes of the tropical mean SWHR responses in the two single-forcing pairs of experiments, ~0.08-0.09 K day$^{-1}$ near the stratopause, are not distinguishable from one another based on the estimated uncertainties and, thus, both effects contribute almost equally to the maximum SWHR anomaly near the stratopause. The RAD-ONLY tropical response
is largest at ~50-60 km, and then decreases sharply with decreasing altitude within the stratosphere. This is related to the intensity of UV radiation being attenuated with increasing path length through the atmosphere. In comparison, the PHOT-ONLY response is smaller than in RAD-ONLY above ~60 km (not shown; see also top of Fig. 2a) but significantly larger in the mid-stratosphere (e.g. by a factor of two at ~40 km). This is due to the SMAX-SMIN increase in tropical ozone in PHOT-ONLY that maximises in the mid-stratosphere (~37 km, Fig. 2c). Thus, while the contributions from the photolysis
and radiation schemes to the SWHR changes are similar near the stratopause, the impact of the enhanced photochemical production of ozone dominates in the mid-stratosphere (in agreement with Shibata and Kodera, 2005, and SPARC, 2010.

The tropical mean SWHR response in INTERO3 reaches up to ~0.16 K day$^{-1}$, and mostly follows the sum of PHOT-ONLY and RAD-ONLY (green line in Figure 2a). Thus, in the tropics, the individual SWHR responses in the single forcing
experiments can be added linearly to give an estimate very close to the full response.

## 4.2. Temperature

The corresponding SMAX-SMIN tropical average temperature responses are shown in Fig. 2b (where $\Delta$TSI = 1.06 Wm$^{-2}$). In INTERO3, the maximum temperature response peaks at ~0.6 K over a fairly broad layer spanning ~45-60 km. Noteworthy, despite the identical implementation of the 11-year solar cycle forcing in the model, the maximum response simulated in
these timeslice runs is somewhat smaller than the response found in the analogous transient UM-UKCA integrations discussed in Bednarz et al. (2019, ~0.8 K/Wm$^{-2}$), likely indicating some contributions of indirect dynamical processes and/or interannual variability to one or both responses. In both cases, the UM-UKCA simulated temperature response is somewhat smaller than found in some reanalyses (e.g. Mitchell et al., 2015b; Bednarz et al., 2019); this could be due to large uncertainties in the responses diagnosed from reanalyses and/or some deficiencies in the model implementation of the solar
cycle forcing (see Bednarz et al., 2019, for details).

Our integrations show significant SMAX-SMIN changes in the upper stratospheric temperatures in RAD-ONLY and PHOT-ONLY, illustrating that the solar cycle impacts on both atmospheric heating and photolysis are important in determining the temperature response there. As noted earlier, there is a large spread in the simulated upper stratospheric temperature responses to the 11-year solar cycle forcing among different atmospheric models (e.g.: Austin et al., 2008; SPARC, 2010; Mitchell et al. 2015a; Hood et al., 2015). Thus, details of both schemes in models and their implementation of the solar cycle forcing can have a strong influence on the simulated stratospheric temperature response to the 11-year solar cycle, and thus to contribute to the inter-model spread.

The estimated standard errors in the magnitude of the temperature responses are comparatively larger than those found for the SWHRs, presumably owing to the additional contribution from dynamical processes to the stratospheric temperature variability through adiabatic heating/cooling. Thus, the temperature responses in RAD-ONLY and PHOT-ONLY are statistically indistinguishable throughout most of the stratosphere. We note that although PHOT-ONLY shows a somewhat stronger SWHR response in the upper stratosphere than RAD-ONLY (Fig. 2a), the associated PHOT-ONLY temperature response there is smaller (Fig. 2b). This illustrates that the atmospheric temperature response to the amplitude of the 11-year solar cycle forcing is not only controlled by changes in SWHRs, but also reflects the associated changes in the longwave component as well as any indirect changes in the circulation (not shown). As discussed above, the combined RAD+PHOT stratospheric temperature response in the tropics is in good agreement with the results from INTERO3 (consistent with Shibata and Kodera, 2005, Gray et al., 2009, and Swartz et al., 2012).

## 4.3. Ozone

Figure 2c shows the simulated changes in the tropical mean ozone mixing ratios. In RAD-ONLY, we find a small SMAX-SMIN ozone decrease (up to ~0.5 %) in the mid-to-upper stratosphere and lower mesosphere. This results from the enhancement of chemical ozone loss under increased temperature, most importantly through the Chapman and $NO_x$ ozone loss cycles (Fig. S1, Supplement, with the change in ozone loss via the Chapman cycle being a factor of ~1.5-6 larger between 40-50 km than via the $NO_x$ cycle; see also e.g., Barnett et al., 1975; Haigh and Pyle, 1982; Jonsson et al., 2004). In contrast, ozone increases in PHOT-ONLY throughout most of the stratosphere and lower mesosphere. This occurs primarily due to the enhanced photolysis of oxygen at wavelengths shorter than ~242 nm (Eq. 1) and the subsequent formation of ozone (Eq. 2), but is also influenced by a solar-induced reduction in the stratospheric $NO_x$ levels (not shown), likely related to its enhanced photochemical removal (e.g. Soukhodolov et al., 2016). The maximum tropical mean stratospheric ozone response in PHOT-ONLY (~3%) is somewhat larger than in INTERO3 (~2.5 %.), reflecting the inverse dependence of ozone on the associated temperature changes (with the temperature-induced modulation of the $NO_x$ cycle playing the dominant role in the mid-stratosphere, Fig. S1, Supplementary Information, see also Jonsson et al., 2004). Throughout most of the tropics, the yearly mean RAD+PHOT ozone response is in a reasonable agreement with the response in INTERO3 (in agreement with Swartz et al., 2012). There is some overestimation of the summed response compared with the control case;

this illustrates that stratospheric ozone concentrations are controlled by a range of photochemical processes, thereby resulting in a complex dependence of the SMAX-SMIN ozone response on the associated temperatures, incoming wavelength-dependent solar radiation as well as any resulting changes in ozone columns above.

To summarise, in the tropics the SMAX-SMIN changes in the SWHRs, temperature and ozone in PHOT-ONLY and RAD-ONLY, which include the solar cycle forcing only in the photolysis and radiation schemes, respectively, can be summed linearly to give a response that is in a good agreement with the full response in the control INTERO3 pair. Our UM-UKCA results agree with the previous FDH calculations of Shibata and Kodera (2005), Gray et al (2009) and SPARC (2010) as well as with the CCM results of Swartz et al. (2012). However, as noted in Sect. 3, the results show larger differences between the
combined and the control temperature responses at high Southern latitudes (Fig. 1d). The following section analyses the corresponding responses modelled during the SH winter and spring, where the role of dynamical processes in modulating the response to solar cycle forcing has been shown to be important (Kuroda and Kodera, 2002; Kodera and Kuroda, 2002).

## 5. The seasonal response in the Southern Hemisphere

The mechanism proposed by Kuroda and Kodera (2002) and Kodera and Kuroda (2002) (thereafter referred to as KK2002a
and KK2002b) to explain the dynamical response to the 11-year solar cycle forcing they identified in reanalysis data postulates that solar-induced changes in the tropical SWHRs and temperatures initiate a chain of feedbacks that modulates the strength of the polar vortex during the dynamically active season. The UM-UKCA simulated changes in zonal mean zonal wind and temperature during SH winter (June-August, JJA) and spring (September-November, SON) for the three pairs of experiments are shown in Figs. 3 and 4, respectively.

The SMAX-SMIN differences in zonal mean zonal wind modelled in the SH high latitudes in INTERO3 are fairly weak and not highly statistically significant in either winter or spring (panels e-f in Figs. 3-4). There is a suggestion of a weak (~0.5 m s$^{-1}$) strengthening of the polar vortex near the stratopause during winter, consistent with the strengthened horizontal temperature gradient. In comparison, the reanalysis data suggest a strengthening of the SH polar jet on its equatorward side
and weakening on its poleward side in winter; this spatial pattern is followed by an enhanced weakening/warming of the vortex in austral spring (e.g.: KK2002a; KK2002b; Frame and Gray, 2010; Mitchell et al., 2015b; Kodera et al., 2016). The disagreement between the model results and reanalysis data could be due to a number of factors, including: i) the uncertainties in the reanalysis SH response; ii) differences between the timeslice runs here with prescribed climatological SSTs/sea-ice and a transient evolution of the real atmosphere and its coupling to the oceans; iii) a positive bias in the model
SH zonal wind climatology (not shown), which may affect interactions between planetary waves and the mean flow.

In RAD-ONLY, the zonal mean SH zonal winds in winter strengthen between SMAX and SMIN on the equatorward flank of the stratospheric/lower mesospheric jet by up to ~3 m s$^{-1}$ (Fig. 3a). This is associated with a cooling of the high latitude stratosphere by up to ~0.75 K (Fig. 4a). The strengthening of the polar vortex in the mid-latitudes extends down to the extratropical troposphere, where it is accompanied by a small (~0.5 m s$^{-1}$) negative zonal wind anomaly in the subtropical troposphere. The latter is indicative of a small poleward shift in the mid-latitude eddy-driven jet (Haigh et al., 2005; Simpson et al., 2009). Whilst the modelled stratospheric responses in RAD-ONLY are generally not highly statistically significant, they bear some resemblance to those found in reanalysis studies (e.g.: KK2002a; KK2002b; Frame and Gray, 2010; Hood et al., 2015; Mitchell et al., 2015b; Kodera et al., 2016). No significant high latitude response was simulated in RAD-ONLY in austral spring (panel b in Figs. 3-4).

In contrast, in PHOT-ONLY there is a strengthening of the stratospheric jet on its poleward side (up to ~1 m s$^{-1}$) and a weakening on its equatorward side (up to ~2.5 m s$^{-1}$) during SH winter (Figs. 3c and 4c). This represents a poleward contraction of the polar vortex, and is accompanied by a warming in the mid-to-upper high latitude stratosphere of up to ~1 K. Importantly, the easterly zonal wind anomaly develops with time, with significantly weaker zonal wind (up to ~3.5 m s$^{-1}$) simulated in the SH mid-to-high latitude upper stratosphere and lower mesosphere in spring (Fig. 3d). Coincident with the zonal wind changes, the Antarctic stratosphere is warmer by up to ~2 K in the austral spring (SON) mean (Fig. 4d). This modulation of the polar vortex persists until the vortex breaks up. A histogram showing the interannual variability of the mid-latitude zonal winds in August simulated in all runs is shown in Fig. S2, Supplement.

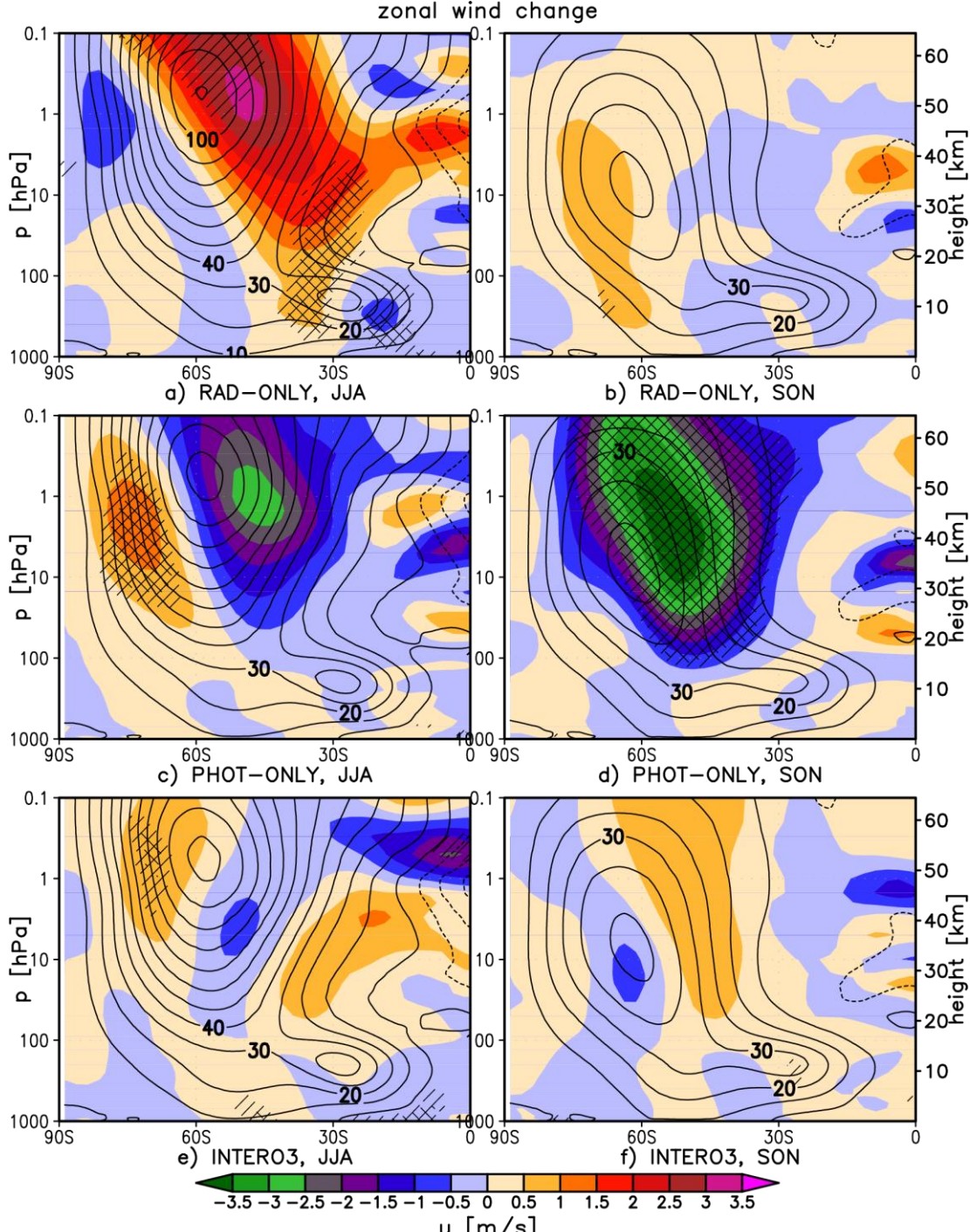

**Figure 3.** Shading: seasonal mean (left: JJA and right: SON) SH zonal mean zonal wind change [m s⁻¹] between SMAX and SMIN for (a-b) RAD-ONLY, (c-d) PHOT-ONLY and (e-f) INTERO3. Single and double hatching indicates statistical significance at the 90% and 95% confidence level, respectively (t-test). Contours show the corresponding climatological seasonal mean zonal mean zonal wind for the respective SMIN run; contour spacing is 10 m s⁻¹.

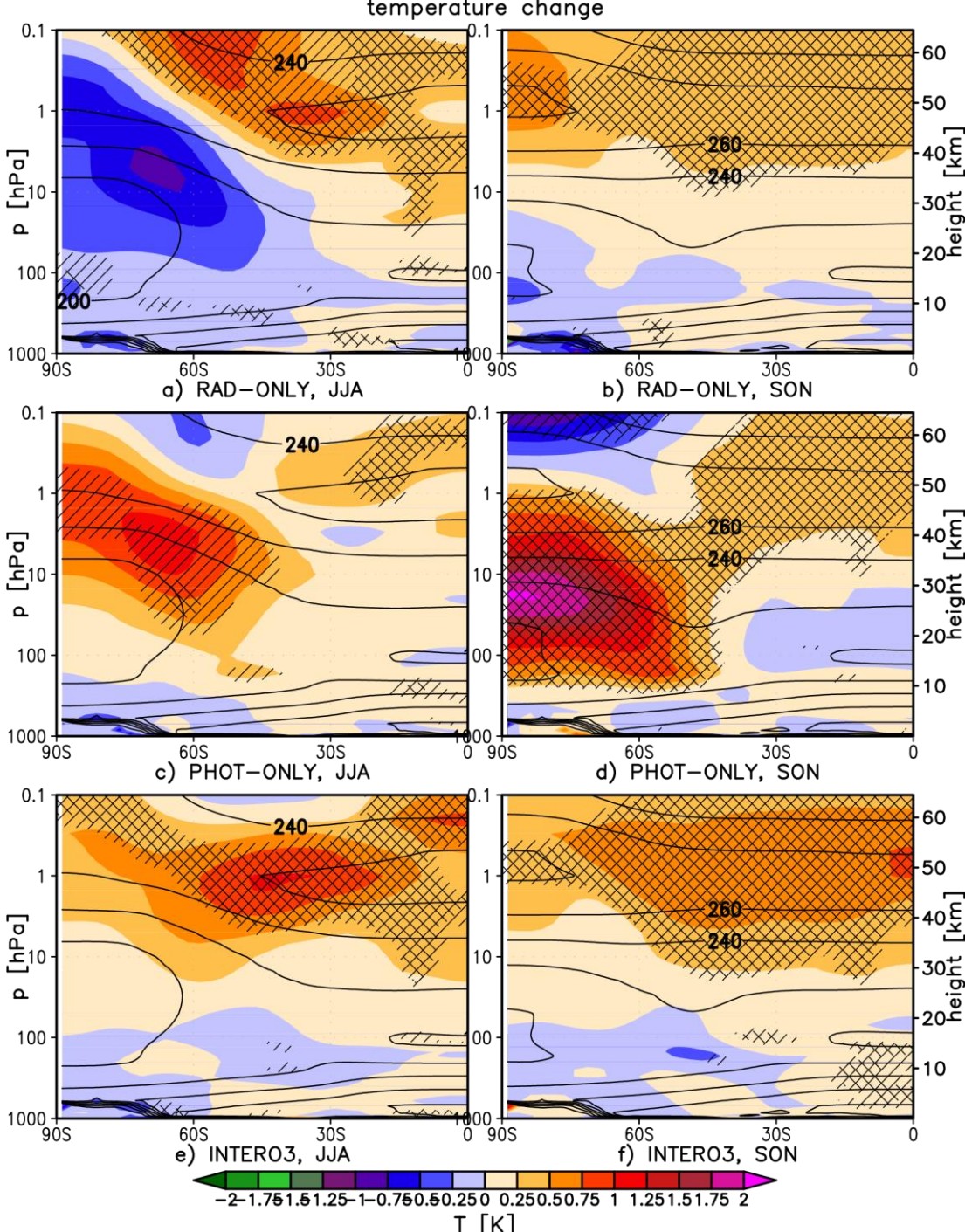

**Figure 4. As in Fig. 3, but for the SMAX-SMIN zonal mean temperature changes [K] (shading) and climatological zonal mean temperature in SMIN run (contours). Contours spacing is 20 K (beginning at 140 K).**

The poleward shift of the stratospheric vortex simulated during winter in PHOT-ONLY and its equatorward strengthening in RAD-ONLY are essentially opposite to one another. Therefore, there is a substantial cancelation between the responses upon linear addition of the JJA means. The combined RAD+PHOT temperature and zonal wind responses in JJA are generally similar to the weak response in INTERO3 (Fig. 5).

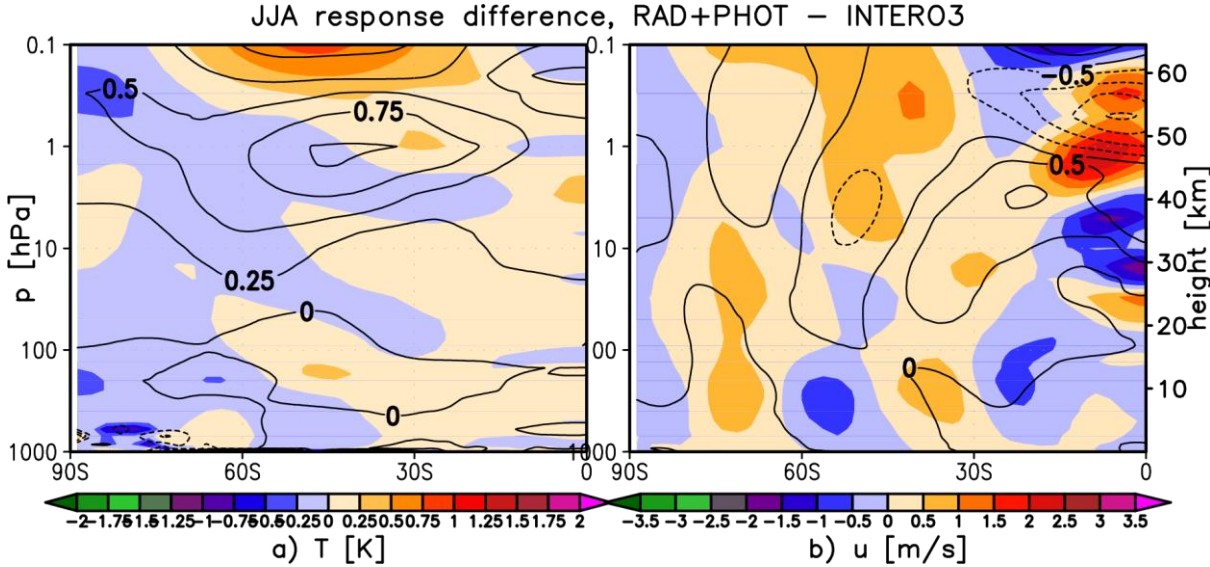

**Figure 5. (a) Shading shows the JJA mean difference between the sum of the RAD-ONLY and PHOT-ONLY temperature [K] responses and INTERO3. (b) as in (a) but for the corresponding zonal wind [ms⁻¹] responses. Contours in (a-b) show the responses in INTERO3 for reference.**

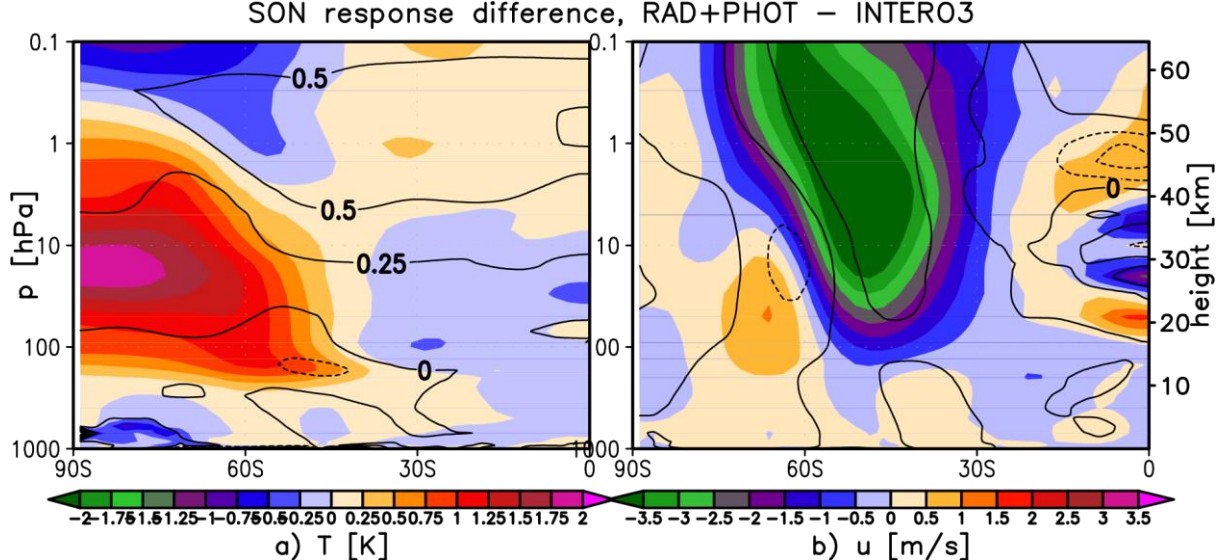

**Figure 6. As in Fig. 5 but for the SON mean.**

In austral spring (SON), a different picture emerges for the comparison between the sum of the PHOT-ONLY and RAD-ONLY responses and the INTERO3 response. In particular, the development of a significantly weaker and warmer polar vortex in PHOT-ONLY in spring contrasts strongly with the small circulation changes found in RAD-ONLY and INTERO3. Consequently, the combined RAD+PHOT responses in SON show larger differences compared to INTERO3 (Fig. 6). There

is a difference in polar temperatures of up to ~1.75 K between the summed and INTERO3 responses, which is large enough to exceed the ±2 standard error confidence interval around the INTERO3 response and be evident in the annual mean (Fig. 1d). We note, however, that this SON difference between RAD+PHOT and INTERO3 is not significant in a strict statistical sense where the confidence intervals around both responses are considered. Nonetheless, our UM-UKCA results highlight that stratospheric high latitude dynamical responses to the amplitude of the 11-year solar cycle forcing are complex and

could be non-additive. We explore this behaviour next.

## 6. Proposed mechanism for the non-linear SH springtime response

The mechanism for an 11-year solar cycle modulation of the polar vortex proposed by KK2002a/b centres on the direct solar-induced warming in the tropical region in autumn/early winter and the immediate changes in the horizontal temperature gradients as the primary driver of the chain of feedbacks between planetary waves and the mean circulation throughout the

winter. To understand the potential reasons for the different dynamical responses simulated among our UM-UKCA experiments, we focus here on the changes in the SWHRs, the primary driver of the anomalous temperature tendencies. We use a simple measure for the solar-induced changes in the horizontal SWHR gradient across the SH as given by Eq. 4:

$$\Delta_{SMAX-SMIN} SWHR_{gradient} = \Delta_{SMAX-SMIN} SWHR_{0-60°S} - \Delta_{SMAX-SMIN} SWHR_{60°S-90°S} \quad (4)$$

### 6.1. SH spring

First, we look at the reasons behind the non-linear springtime response. The original mechanism proposed by KK2002a/b considers only the direct solar-induced temperature changes in the tropics during autumn/early winter as the primary driver of the high latitude circulation responses throughout the dynamically active season. However, our results suggest that changes in the SWHR gradients throughout the whole time period are important for the evolution of the SH dynamical response. During spring it is the SWHR changes at higher latitudes, influenced strongly by the changes in ozone, that can be

particularly important for determining the horizontal gradients owing to the increasingly higher mean insolation following the onset of spring.

In particular, the springtime changes in the SWHR horizontal gradients near 60°S in RAD-ONLY and INTERO3 (Fig. 7b) have similar vertical structure and both are much smaller than their corresponding gradient changes in winter (Fig. 7a).

These small SON horizontal gradient changes, arising from the similarity between SWHR responses in the tropics/mid-latitudes and the polar regions (Fig. S3b,d, Supplementary Information), give rise to small zonal wind and temperature

responses in the two pairs of experiments. In stark contrast to this, PHOT-ONLY shows a markedly different SWHR gradient change (Fig. 7b): while the gradient strengthens substantially at ~40 km, the response is negative in both the lower mesosphere as well as in the lower stratosphere. The response is dominated by the strong contribution of the high latitude SWHR response, which shows an alternating positive and negative pattern (Fig. S3d).

These high latitude SWHR changes are strongly related to the changes in polar ozone (Fig. 9). We find that ozone mixing ratios in PHOT-ONLY increase in winter and spring not only in the tropics but also throughout large parts of the polar stratosphere (Fig. 9c-d). In fact the percentage changes in polar ozone, in particular during spring, can be larger than those in the tropical/mid-latitude stratosphere. These are likely to occur due to a combination of elevated ozone levels already locally

present before the start of the dynamically active season (not shown) and changes in the circulation/transport. In line with the simulated enhancement of the stratospheric meridional circulation (Fig. 10) and, thus, increased transport of ozone-rich air from the tropics and higher polar altitudes, ozone anomalies are transported poleward and downward; the percentage ozone anomalies also appear to magnify in spring. Further feedbacks may also be possible due to any resulting coupling with temperature and/or chemical loss cycles that may follow (e.g.: Hood et al., 2015). As more solar radiation reaches the high

latitudes in late winter/spring, any changes in ozone there become increasingly important for determining the horizontal SWHR gradients and, hence, for feeding back and modulating the mean flow. This marked pattern of changes in SWHR gradients in PHOT-ONLY accompanies comparatively larger zonal wind and temperature responses, Figs. 3 and 4. The schematic representation of such mechanism is in Fig. 8b,d.

### 6.2. SH winter

Secondly, we consider why the two single-forcing experiment pairs (RAD-ONLY and PHOT-ONLY) indicate contrasting SH winter polar vortex responses. We find that during winter (JJA), the maximum changes in the SWHR gradient near 60°S (Fig. 7a) in our runs peak at different altitudes, with the strongest changes in gradient found in the lower mesosphere in RAD-ONLY and in the upper stratosphere in PHOT-ONLY. Little insolation reaches the SH high latitudes in winter, and thus the SWHR responses there are small (Fig. S3c, Supplementary Information), so that the changes in the horizontal

gradients in winter are dominated by the SWHR responses in the SH tropics/mid-latitudes (Fig. S3a). The latter are largely similar to those found for the tropical annual mean in Fig. 2a, following the same arguments as in Sect. 4.1. We also find that the development of the SH zonal wind and temperature anomalies in our experiment pairs is associated with changes in planetary wave propagation and breaking: the wave propagation/breaking is increased in PHOT-ONLY and reduced in RAD-ONLY, with no well-defined changes in INTERO3 (Fig. S5 and S6, Supplementary Information). To our knowledge

few studies have examined the role of the spatial structure of the anomalous solar-induced tropical temperature tendencies for the resulting high latitude dynamical response (e.g. Ito et al., 2009, who looked at horizontal structure). Possibly, the propagation and breaking of planetary waves within the stratosphere may be sensitive to the spatial, in our case the vertical, structure of the anomalous SWHRs. These would act to alter temperature tendencies, thereby influencing zonal winds and

potential vorticity gradients that are important for planetary wave propagation. The details of such potential sensitivity are, however, difficult to diagnose using our experiments and this hypothesis requires further examination with additional sensitivity experiments. Another potential reason for the differences in the simulated winter responses between the integrations may be the role of zonally asymmetric ozone heating in influencing planetary wave propagation. Numerous

5    studies have shown that stratospheric ozone, as a radiative gas, can influence planetary wave propagation, thereby impacting on the interaction between the planetary waves and mean flow (e.g. Nathan and Cordero, et al., 2007, Kuroda et al., 2007, 2008; McCormack et al., 2011, Silverman et al., 2018). Possibly, the presence of increased ozone levels in PHOT-ONLY may act in a similar manner, enhancing the impact of such zonally asymmetric ozone heating. As before, this hypothesis should be subject to further testing. The schematic representation of the proposed mechanism is shown in Fig. 8a,c.

The sum of the changes in the SWHR gradients in RAD-ONLY and PHOT-ONLY agrees with INTERO3 in austral winter. However, this agreement is not found in austral spring: the sum of the single-forcing responses is dominated by the changes in PHOT-ONLY and is not additive, consistent with the lack of additivity of the zonal wind and temperature responses in SON (Fig. 6). Therefore, our results highlight the need to implement the solar cycle forcing interactively in both the radiative

15   heating and photolysis schemes to fully capture the complex feedbacks between the photochemistry, radiation and dynamics.

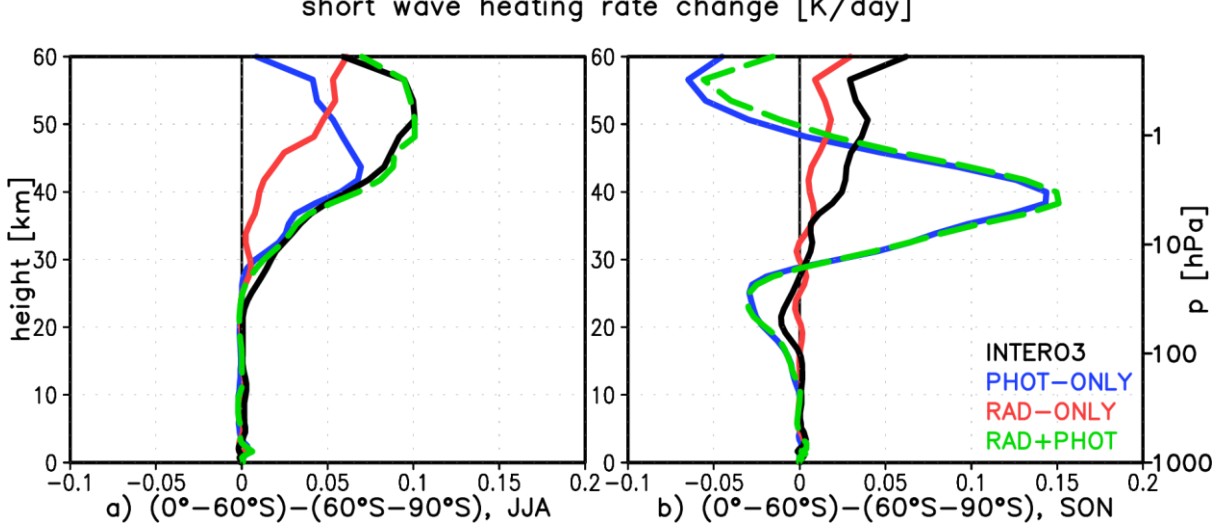

**Figure 7. Seasonal mean JJA (a) and SON (b) change in the SWHR gradient [K day⁻¹], as defined in Eq. (4), between SMAX and SMIN for INTERO3 (black), PHOT-ONLY (blue), RAD-ONLY (red), and PHOT-ONLY + RAD-ONLY (green).**

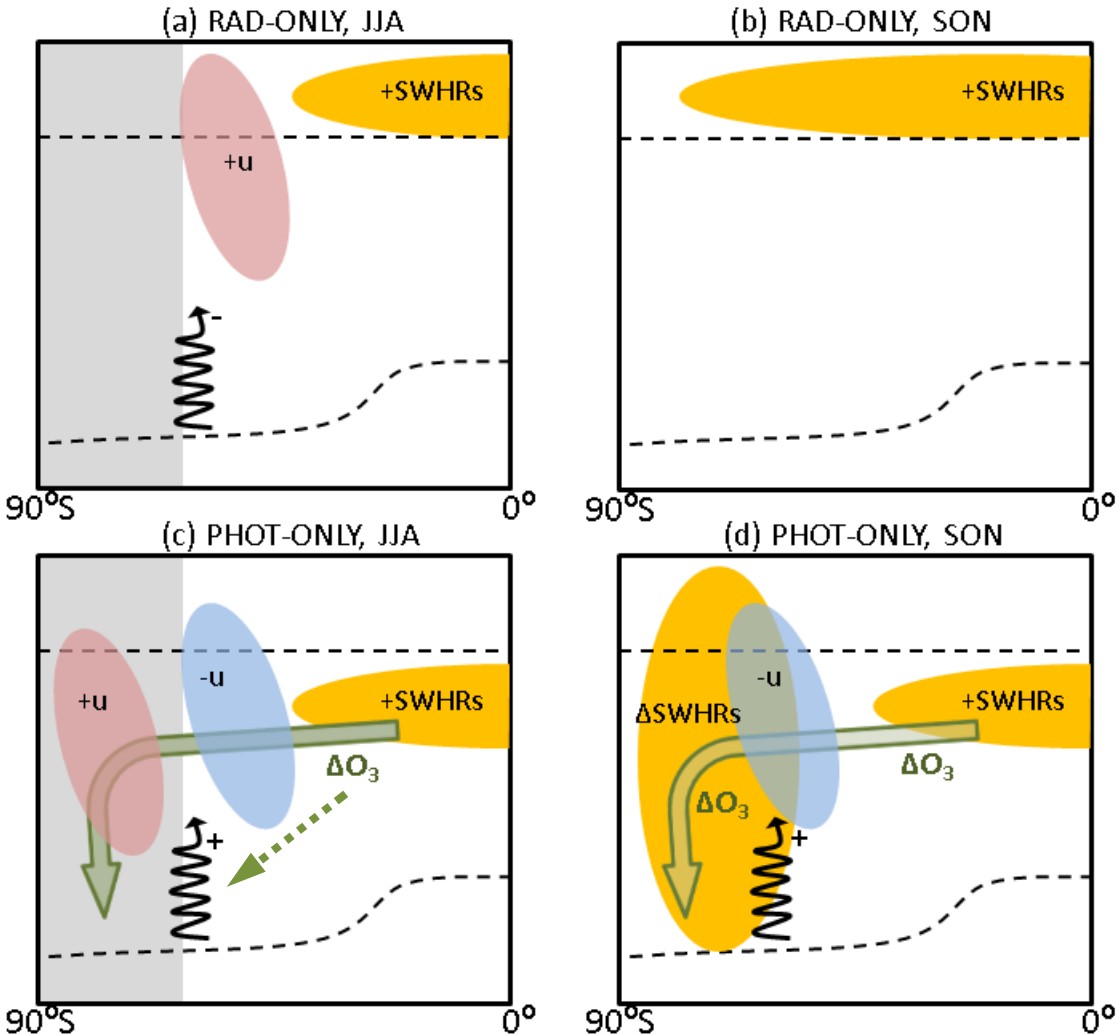

**Figure 8. Schematic representation of the proposed mechanism. Yellow ovals represent changes to the SWHR, red and blue ovals represent strengthening and weakening of zonal mean zonal wind, respectively. The green arrow indicates changes in ozone along the meridional circulation, the wavy black arrows the propagation of planetary waves (increased/decreased as given by the plus/minus signs), and the dotted green line an interaction between ozone and planetary waves. The dashed horizontal lines indicate tropopause and stratopause, and the grey areas in (a,c) the regions covered in polar night.**

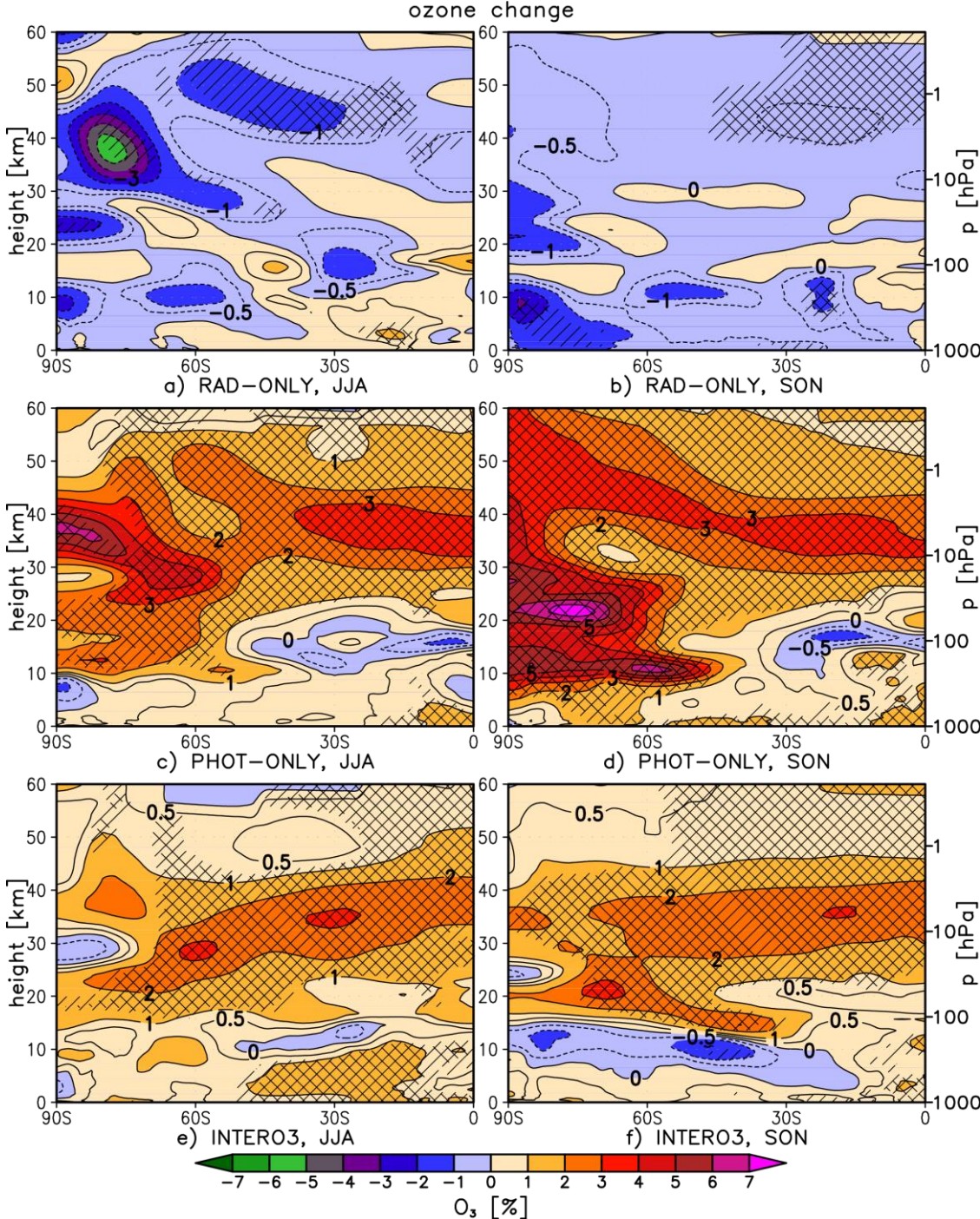

**Figure 9.** Seasonal mean (left: JJA, right: SON) SH zonal mean changes in ozone mixing ratios [%] between SMAX and SMIN for (a-b) RAD-ONLY, (c-d) PHOT-ONLY and (e-f) INTERO3. Single and double hatching indicates statistical significance at the 90% and 95% confidence level, respectively (t-test) Note the additional contours at ±0.5 %.

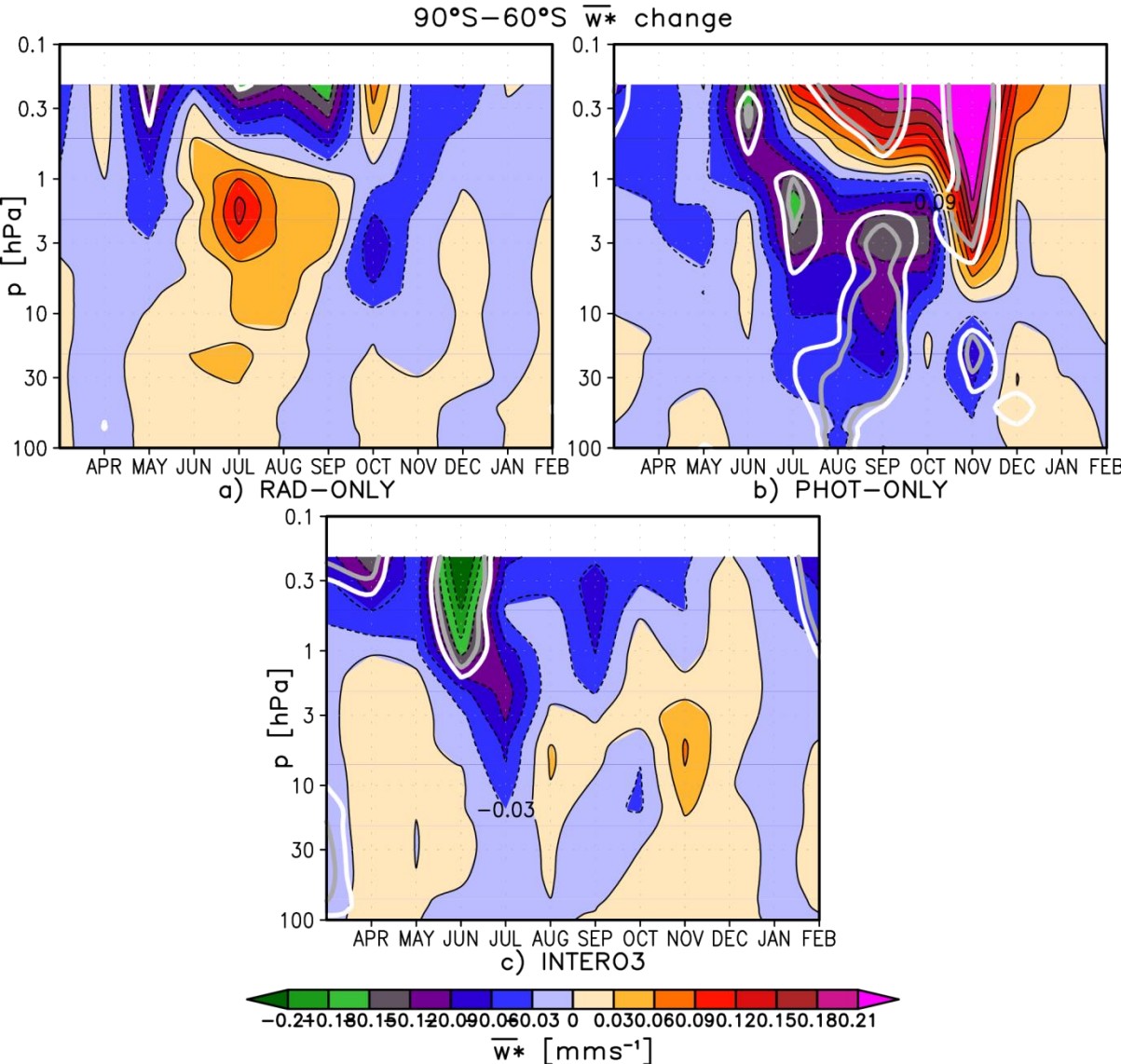

**Figure 10. The monthly mean evolution of the polar cap average (90°S-60°S) change in the vertical component of the Transformed Eulerian mean circulation, $\bar{w}^*$ [mm s⁻¹], between SMAX and SMIN for (a) RAD-ONLY, (b) PHOT-ONLY and (c) INTERO3. Positive values indicate anomalous upwelling, and vice-versa. Thick white and grey lines indicate statistical significance at the 90% and 95% confidence level, respectively.**

## 7. Discussion

Haigh (2010) pointed out that the solar cycle induced ozone response alters the penetration of solar radiation to lower altitudes and, therefore, leads to a stratospheric SWHRs response that is complex and non-linear, depending on the associated changes in ozone. In agreement, our results indicate that the changes in ozone associated with photochemical production and coupling to the circulation, not only in the tropics/mid-latitudes but also in the polar regions, are important

for modulating the SH dynamical response to the amplitude of the 11-year solar cycle. A similar conclusion was reached by Hood et al. (2015) who suggested that both increases in tropical ozone as well as dynamically-induced sharp horizontal ozone gradients at higher latitudes are important for horizontal temperature gradients in the stratosphere and thus play a role in amplifying the associated seasonal zonal wind response. In addition, Kuroda and Kodera (2005), Kuroda and Shibata (2006) and Kuroda et al. (2008) found that under higher solar activity any changes in polar ozone driven by the Brewer-Dobson circulation during winter can persist in the lower stratosphere for several months, thereby inducing local temperature and zonal wind responses. In agreement, while the dynamical response simulated in RAD-ONLY largely disappears by spring, the response in PHOT-ONLY develops with time, with changes in polar ozone potentially contributing to this behaviour.

The importance of springtime high latitude ozone changes in modulating the SH polar vortex has also been recognised in the context of halogen-induced Antarctic ozone depletion (e.g.: McLandress et al., 2010; Keeble et al., 2014). There have also been indications that the role of ozone, as a radiatively active gas, is important in influencing the interactions between planetary waves and the mean flow, thus modulating the dynamical response to the solar cycle forcing (e.g.: Kuroda et al., 2007; 2008; Nathan and Cordero, 2007). McCormack et al., 2011, showed that inclusion of zonally-asymmetric ozone heating in their model weakens the climatological winter NH polar vortex. The idea that increased ozone levels at SMAX may act in a similar manner has been proposed by other studies (e.g. Kuroda et al., 2007; 2008; Nathan and Cordero, 2007), although the importance of this effect for the solar SH dynamical response has been recently questioned (Kuroda and Deushi, 2016).

Hood et al. (2015) argued that it is important that models reproduce the significant ozone and temperature responses that have been observed in the tropical upper stratosphere in order to simulate stronger amplification of the horizontal temperature gradients at these altitudes. A comparison with the altitude differences between the changes in the SWHR gradients found in RAD-ONLY and PHOT-ONLY in winter raises an interesting question of whether the SH dynamical response could be sensitive not only to the magnitude of the changes in the SWHR gradient but also to its maximum altitude range. It is now accepted that variability in the tropical stratosphere can affect the high latitudes due to its impact on the planetary wave propagation and breaking. For instance, a number of studies reported evidence for the influence of the Quasi-Biannual Oscillation (QBO) on the polar vortex or on the development of the NH high latitude dynamical response to the solar cycle forcing (e.g. Holton and Tan, 1980; Labitzke et al., 2006; Ito et al., 2009; Matthes et al., 2013; Watson and Gray, 2015). Assuming that changes in the zonal momentum forcing associated with the different phases of the QBO modulate the vertical structure of the tropical temperatures, then a similar mechanism involving changes in wave-mean flow interactions might operate here, although specially designed experiments would be required to further diagnose the details of these sensitivities.

All in all, the apparent non-additive character of the dynamical response simulated in our experiments during the SH spring argues strongly for the need to include the solar cycle forcing interactively in both the radiation and photolysis schemes in order to fully capture the atmospheric response to the 11-year solar cycle.

## 8. Conclusions

The atmospheric response to the amplitude of the 11-year solar cycle forcing in the UM-UKCA chemistry-climate model has been separated into the contributions resulting from direct radiative heating and from changes in photolysis. Pairs of sensitivity timeslice experiments representing maximum and minimum conditions of the 11-year solar cycle were performed with the solar cycle forcing included exclusively in either the model radiation or photolysis scheme. The sum of the two single-forcing responses was compared with a control pair with both effects included.

In the tropical upper stratosphere, the yearly mean SMAX-SMIN shortwave heating rate responses in the radiation-only and photolysis-only experiments were found to be of similar magnitudes, with both resulting in significant temperature responses near the stratopause. Details of the implementation of the solar cycle forcing in the individual schemes in models will have an important influence on the simulated tropical stratospheric temperature responses to the solar cycle forcing. Hence, this will be important when considering the large inter-model spread in the atmospheric response to the 11-year solar cycle forcing reported in the literature. Below the stratopause, the shortwave heating anomaly in the radiation-only case decreases sharply with decreasing altitude and is smaller than in the photolysis-only experiment. However, the corresponding upper stratospheric temperature response is ~0.1 K larger, illustrating that the stratospheric temperature response to the amplitude of the solar cycle forcing is not just the result of the shortwave heating rates perturbation but is also influenced by any changes in the longwave component as well as any indirect dynamical processes (Section 4.2). For ozone, the radiation-only case shows a small (~0.5 %) decrease in the tropical upper stratospheric ozone at SMAX due to the acceleration of chemical ozone loss at higher temperatures. In contrast, in the photolysis-only case tropical ozone abundances increase by up to ~3 % due to the enhanced $O_2$ photolysis and the subsequent ozone production. The magnitude of the tropical stratospheric ozone response in the photolysis-only case is slightly larger than in the control case, in line with the inverse dependence of ozone concentrations on temperature.

The pairs of experiments showed different SH high latitude circulation responses between the 11-year solar cycle maximum and minimum in austral winter and spring. In the radiation-only case, the stratospheric responses at high southern latitudes were not highly statistically significant, but the results suggest a strengthening of the polar vortex during winter on its equatorial side and a cooling of the polar stratosphere at solar maximum broadly consistent with the reanalysis. In contrast, in the photolysis-only we find a poleward contraction of the polar vortex and an associated warming of the polar stratosphere. In JJA, the sum of these two distinct responses shows strong cancellation and compares well with the small

vortex response in the case including both radiation and photolysis effects together. However, this agreement was not found in austral spring (SON), where the springtime weakening and warming of the polar vortex found in the photolysis-only case is in stark contrast with the negligible responses in the other simulations.

In order to understand a mechanism behind the different dynamical behaviour in our runs and the resulting non-linear springtime response, an analysis of the corresponding shortwave heating rate gradients across the Southern Hemisphere was performed. We find differences in the magnitude and vertical structure of their changes in winter. This raises a question about a potential sensitivity of the dynamical response to the altitude of the anomalous radiative tendencies, although this hypothesis requires further testing. Another potential factor contributing to the different winter responses may be the role of
enhanced zonally-asymmetric ozone heating brought about by the increased ozone levels in modulating planetary wave propagation and breaking, Our results thus act as a motivation for further study. Importantly, we find marked changes in the Antarctic shortwave heating rates in the photolysis-only case in spring; these make a strong contribution to the associated changes in the horizontal shortwave heating rate gradients. These high latitude changes are predominantly driven by the photochemical ozone changes and their coupling to the circulation changes (Fig. 10), but further feedbacks due to any
resulting coupling with temperatures/chemical loss cycles could also play a role. As changes in the horizontal shortwave heating rates gradients throughout the dynamically active season could feed back on and modulate the mean flow, this is a plausible mechanism to explain the simulated weakening and warming of the polar vortex in spring.

All in all, the tropical yearly mean shortwave heating rates, temperature and ozone responses in both the photolysis-only and
the radiation-only cases are found to be important for determining the full direct stratospheric response to the amplitude of the 11-year solar cycle forcing, with both effects being largely, albeit not fully, additive in the tropics. However, the apparent non-additive character of the high latitude dynamical responses simulated in the SH spring strongly argues for the need to include the solar cycle forcing interactively in both the radiation and photolysis schemes in order to capture the complex feedbacks between photochemistry, dynamics and radiation and, thus, in order to fully model the atmospheric response to the
11-year solar cycle forcing.

**Acknowledgments**

We acknowledge funding from the ERC for the ACCI project grant number 267760, including PhD studentship for EMB. ACM, JAP, PJT and NLA were supported by the National Centre for Atmospheric Science, a NERC funded research centre.
ACM acknowledges support from an AXA Postdoctoral Fellowship and a NERC Independent Research Fellowship (NE/M018199/1). We acknowledge the use of HECToR, the UK's national high-performance computing service. The authors thank Remi Thiéblemont and one anonymous reviewer for helpful comments that have improved the manuscript.

Authors contributions:

EMB run the model experiments, analysed the data and wrote the paper, with discussion, feedback and input from ACM, PB and JAP. NLA provided the model version and PJT implemented the 11-year solar cycle forcing into the model.

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
