# Peer review of "Separating the role of direct radiative heating and photolysis in modulating the atmospheric response to the amplitude of the 11-year solar cycle forcing"

_Atmospheric Chemistry and Physics, 2018_

## Referee Comment (RC1) · R. Thiéblemont (Referee) · 26 Apr 2018

Review of the manuscript "Separating the role of direct radiative heating and photolysis in modulating the atmospheric response to the 11-year solar cycle forcing" by Bednarz et al.

Reviewer: Rémi Thiéblemont

This paper examines the thermal and dynamical responses of the tropical and Southern Hemisphere polar stratosphere to changes in solar irradiance using sensitivity experiments of the chemistry-climate model UM-UKCA. The aim of the paper is to separate the effect of the photochemistry and radiation module (that are artificially separated in models) on the solar signal and to explore the linearity of the stratospheric response due to the photochemistry and radiative module contributions. They found that the response is linearly additive in the tropics but not in the polar region and proposes mechanisms to explain the non-linearity in the polar region.

The issues that the paper is addressing have long been debated and the results of the paper constitute an added value toward a better understanding of the impact of the solar variability on the stratosphere (and thus potentially on climate) but also on the importance of model design on the representation of the solar variability-induced effects. The main findings of this paper are novel and constitute an interesting scientific contribution in my opinion. I find the manuscript also well-structured and well written. However, I have some concerns with some interpretations which seem to me somewhat speculative since I don't find that they are convincingly supported by the results. Some statements should hence be tone down unless additional analysis (or experiment) are carried out. Therefore, some revisions of the paper are needed before I recommend it for publication in ACP. I also have several questions for the authors. Please find the details of my comments below.

**Main comments**:

1/ Given that all results and conclusions of this paper are based on timeslice experiments performed under permanent max or min solar conditions, I do not find that it is appropriate to claim that the study investigates the "… atmospheric response to the 11-year solar cycle forcing" (title of the manuscript). This is misleading for readers and should be formulated differently in the title, the abstract, but also everywhere in the manuscript where required (at several places). It could instead be mentioned that the study investigates the atmospheric response to constant changes in solar forcing that correspond to the amplitude of the 11-year solar cycle. Or something like this. Note however that I fully understand the arguments and agree with the benefit of using timeslice experiments instead of transient experiments in this paper.

2/ Presently, the mechanism that is proposed on Fig 8 is not clear to me. In particular, the paragraph and analysis describing the mechanism in link with the changes in wave activity (P15,L24-P16,L2) seem speculative in my opinion. For instance, the only actual significant signal in the wave activity diagnostics (S3 and S4) is seen during 2 months over the July/August/September period for the PHOT-only experiment. The other experiments do not show statistical evidence of changes. What the analysis reveals is that the PHOT-only experiment shows an increased wave activity entering the stratosphere (S3) and increased westward forcing of the mean flow in the upper stratosphere by wave breaking (S4). This is associated with an acceleration of the stratospheric overturning circulation which brings more ozone to the polar region. This could come from background changes in the stratosphere (due

e.g. to the changes in the SWHR gradient as claimed but that may come also from other processes), but also to changes in the wave excitation in the troposphere (see comment 3/). Attributing these wave changes to the SWHR gradient is to me not yet supported by robust evidences. Although I understand that making additional extended analysis may not be easily feasible or wanted, you may consider examining the monthly evolution of the wave activity (amplitude, propagation, …), Brewer-Dobson circulation, wind, SWHR,… to explore the seasonal march of the signal: such analysis may help identifying more clearly some causality. You may also consider examining the refractive index to see if the SWHR changes affect the propagation conditions of wave. Finally, I think that it may be interesting to examine more in details and possibly show how the inter-annual variability behaves for these various quantities. Are the changes in PHOT-only the result of a few years with an "extreme" behavior - for instance possible SSWs in the Southern Hemisphere - or rather the result of more permanent/continuous changes. As it is claimed that the initial source of perturbations in wave activity start from the changes in the SWHR in winter in the tropical region (where the perturbed vertical profile should not experience too much inter-annual variability), I would expect the changes in winter to be rather continuous. The mechanism in spring is much clearer (more ozone in polar region => changes in SWHR gradient, etc) but largely depend on the winter circulation perturbation that is presently not easy to understand.

3/ In light of my previous comment, I wonder if some of the identified changes between the PHOT-ONLY vs. RAD-ONLY & INTERO3 may not partly come from the fact that (if I understood correctly the experimental design) PHOT-ONLY MAX/MIN pair has a constant TSI-induced heating (since the radiation module solar forcing is fixed) while this is not the case for the two other experiments since between MAX & MIN conditions, the TSI-induced changes are considered in the radiation module. Could that lead to some bottom-up residual influence (even if the SSTs are fixed) and contribute to some of the identified differences? Is there a way to diagnose this? Do you think that this could have an influence?

**Specific comments**:

P1, L21-24. As described in comment 2, I am not convinced yet that the SWHR gradients in winter play an important role.

P2, L26-28. Please give one or two example of "…specific aspects of model design" to make the issue more concrete.

P3, L11-12: Indeed, the SH is less studied than NH, but there are clearly more studies than the one cited here (e.g. Petrick et al. (2012, JGR), see also numerous studies of Yuhji Kuroda and co-authors (the most recent by Kuroda was published this year in JGR)). Please cite some.

P3, L25: the "main" or the "only"?

P3, L21-P4, L7. These 2 § are misleading since they leave the impression that transient simulations are performed. For instance, it is mentioned that HadISST are used and an 11-year forcing is implemented while it's not really the case. The authors should make clear from the beginning that they do idealized experiments that look at SMAX-SMIN conditions of the amplitude of the 11-year solar cycle. Presently, it is too confusing in my opinion.

P4, L23-24 "The third pair, PHOT-ONLY SMAX/SMIN, is analogous to RAD-ONLY SMAX/SMIN, but the solar cycle forcing is included exclusively in the photolysis scheme while

constant TSI and SSI are used in the radiation scheme." As mentioned in the main comment 3/, could the TSI change between SMAX and SMIN in the RAD-ONLY also be responsible for the observed difference in the signals? Would it not have been an option to keep always the TSI constant? Note that this would have also made the use of the same fixed climatological SST prescribed for SMAX and SMIN more adequate in the case of the RAD-ONLY and INTERO3 experiments.

P6-7, section 3 and Figure 1. It would be relevant, I think, to calculate also the statistical significance of the difference between the RAD+PHOT & INTERO3 responses (panel d of Figure 1). This would strengthen the results and help motivating the analysis that are carried out later on in the paper.

P8, L20-24. That is very interesting to notice. Could that be due to the extraction method too used in the case of transient experiments and reanalysis and e.g. the difficulty to separate the solar signal from contributions of other variability factors? (see e.g. Chiodo et al., 2014, ACP)

P9, L15-16. It appears from Figure S1 that Chapman dominates over NOx. This could be clarified in the text by giving the contribution of each e.g. in %.

P9, L27. The "small overestimation" is also statistically significant at the 2-sigma level near the peak at ~36 km. That means that even in the tropic, the RAD+PHOT contributions are not exactly linearly additive. This is I think still important to highlight and it shows that the complexity of the system needs to be accounted for, even in regions where we usually believe that the response is simple. Of course it's not major, but still worth mentioning.

P10, L19-23. What about the comparison with the results of Bednarz et al., 2018? Does the comparison between the timeslice and transient experiments help to get further understanding of the opened issues listed here? As you refer to this comparison previously in the manuscript (P8, L20-24), it may be worth looking at it again here.

P10, L26-30. It may be relevant to add the climatologies to the plots (as black contours on the background similarly to Fig. 1d). That would help to better visualize the jet strengthening and eddy driven jet displacement.

P11, L4-7. Did you also look at the inter-annual variability? Is there a difference between the different runs? Are there SSWs–like perturbations (even though they should be rare in SH) that may be responsible for the SH easterly anomalies of the PHOT-Only experiment?

Figures 5 and 6. Similarly to Figure 1d, I think that the statistical significance of the differences could be relevant to show here.

P15, L19-20. Please indicate the altitudes of the peak (lower mesosphere, upper stratosphere is somehow vague). Where in the mesosphere does SWHR peak in RAD-Only (is 60 km the maximum?)?

P15,L24-P16,L2 & schematic on Fig. 8. As explained in comment 2/, that paragraph is not clear and too speculative to me. Please consider either making further analysis to support the present discussion or just tone down.

P16,L6. Instead of "primary driver", I would rather say that it's considered as the "initial driver".

P16, L32-P17,L3. The arguments in this paragraph are not really convincing to me, since despite the fact that SWHR gradient are additive in JJA, the temperature and wind responses are not.

---

## Referee Comment (RC2) · Anonymous Referee #2 · 27 Apr 2018

The manuscript attempts to disentangle the contributions from direct radiation and chemical effects of the solar irradiance. The authors applied the chemistry-climate model (CCM) UM-UKCA to simulate the steady-state atmospheric state for the solar maximum and minimum conditions. To separate the role of different processes and estimate the linearity of the overall response, the authors performed three pairs of experiments switching on only the direct influence of solar irradiance on radiation (RAD-ONLY), photolysis (PHOT-ONLY) and both (INTERO3). Several similar studies have been published before. The novelty of the manuscript consists of the application

of more sophisticated model and the results concerning the non-linearity of the two considered processes during the southern hemisphere cold season. This conclusion is important for the community because it emphasizes the necessity of the interactive ozone treatment in the climate models. However, this important conclusion is also a weakest part of the manuscript because the identification of the responsible mechanisms is not convincing. This issue should be clarified before the publication of the manuscript, otherwise this important conclusion will not be fully appreciated.

Major issues

1. The chain of physical/chemical processes leading to the weaker polar vortex during SON is not convincingly presented. From the presented results, it is more or less clear that the story should start before the winter time. The gradient in heating rate for the PHOT-ONLY case should related to ozone gradient. It cannot be dramatically different from RAD-ONLY case, because the ozone increase inside polar vortex due to enhanced solar UV should not be large. During SON the obtained gradient in ozone is high, but I think it is rather related to dynamical processes during late winter. This dynamically induced increase of the ozone in the stratosphere produce strong heating rate gradients during SON and produce further suppression of polar night jet. Thus, the triggering process is not identified leading to weak understanding of the obtained results. I do not know which process can be involved, but I think the authors should try hard to find it.

2. The linearity of the atmospheric response to radiation and chemical processes was discussed in several previous publications. Maybe it is better to concentrate on discovered no-linearity in the southern hemisphere and makes the description of the annual and tropical mean responses shorter.

Minor issues:

1. Figure 1: Statistical significance is missing on panel d).

2. Section 4.1: I recall the contribution of radiation and ozone effects were analyzed in Forster et al. (2011). Maybe it makes sense to mention this paper?

3. Page 10, Section 5: I think it is pointless to carefully compare the results of time-slice model runs with permanent solar max/min conditions with observations and try to explain the difference.

4. Page 15: The explanations in the last paragraph are too vague and not instructive to my taste. These processes definitely exist, but it is not easy to illustrate how they work.

5. Page 21, line 10: I wonder how it is possible keep this paper under review for already 3 years. It seems something is wrong with it. I would not site unpublished papers, because the results could be wrong.

6. Page 23, line 3: I do not see clear time line of the changes. It looks like triggering mechanism is missing.
* * *

---

## Author Response (AR1)

**RESPONSE TO REVIEWER 1**

This paper examines the thermal and dynamical responses of the tropical and Southern Hemisphere polar stratosphere to changes in solar irradiance using sensitivity experiments of the chemistry-climate model UM-UKCA. The aim of the paper is to separate the effect of the photochemistry and radiation module (that are artificially separated in models) on the solar signal and to explore the linearity of the stratospheric response due to the photochemistry and radiative module contributions. They found that the response is linearly additive in the tropics but not in the polar region and proposes mechanisms to explain the non-linearity in the polar region.

The issues that the paper is addressing have long been debated and the results of the paper constitute an added value toward a better understanding of the impact of the solar variability on the stratosphere (and thus potentially on climate) but also on the importance of model design on the representation of the solar variability-induced effects. The main findings of this paper are novel and constitute an interesting scientific contribution in my opinion. I find the manuscript also well-structured and well written. However, I have some concerns with some interpretations which seem to me somewhat speculative since I don't find that they are convincingly supported by the results. Some statements should hence be tone down unless additional analysis (or experiment) are carried out. Therefore, some revisions of the paper are needed before I recommend it for publication in ACP. I also have several questions for the authors. Please find the details of my comments below.

**We thank the Reviewer for the positive review and constructive comments that have improved the manuscript. Our replies to the individual comments are shown below in blue.**

**Main comments:**
1/ Given that all results and conclusions of this paper are based on timeslice experiments performed under permanent max or min solar conditions, I do not find that it is appropriate to claim that the study investigates the "... atmospheric response to the 11-year solar cycle forcing" (title of the manuscript). This is misleading for readers and should be formulated differently in the title, the abstract, but also everywhere in the manuscript where required (at several places). It could instead be mentioned that the study investigates the atmospheric response to constant changes in solar forcing that correspond to the amplitude of the 11-year solar cycle. Or something like this. Note however that I fully understand the arguments and agree with the benefit of using timeslice experiments instead of transient experiments in this paper.

**We have changed the manuscript (both the abstract and main text) to make it clear that we investigate the response to the amplitude of the 11-year solar cycle forcing using an idealised timeslice setup. We have also changed the title to "… to the amplitude of the 11-year solar cycle forcing".**

2/ Presently, the mechanism that is proposed on Fig 8 is not clear to me. In particular, the paragraph and analysis describing the mechanism in link with the changes in wave activity (P15,L24-P16,L2) seem speculative in my opinion. For instance, the only actual significant signal in the wave activity diagnostics (S3 and S4) is seen during 2 months over the July/August/September period for the PHOT-only experiment. The other experiments do not

show statistical evidence of changes. What the analysis reveals is that the PHOT-only experiment shows an increased wave activity entering the stratosphere (S3) and increased westward forcing of the mean flow in the upper stratosphere by wave breaking (S4). This is associated with an acceleration of the stratospheric overturning circulation which brings more ozone to the polar region. This could come from background changes in the stratosphere (due e.g. to the changes in the SWHR gradient as claimed but that may come also from other processes), but also to changes in the wave excitation in the troposphere (see comment 3/). Attributing these wave changes to the SWHR gradient is to me not yet supported by robust evidences. Although I understand that making additional extended analysis may not be easily feasible or wanted, you may consider examining the monthly evolution of the wave activity (amplitude, propagation, ...), Brewer-Dobson circulation, wind, SWHR,... to explore the seasonal march of the signal: such analysis may help identifying more clearly some causality. You may also consider examining the refractive index to see if the SWHR changes affect the propagation conditions of wave. Finally, I think that it may be interesting to examine more in details and possibly show how the inter-annual variability behaves for these various quantities. Are the changes in PHOT-only the result of a few years with an "extreme" behavior - for instance possible SSWs in the Southern Hemisphere - or rather the result of more permanent/continuous changes. As it is claimed that the initial source of perturbations in wave activity start from the changes in the SWHR in winter in the tropical region (where the perturbed vertical profile should not experience too much inter-annual variability), I would expect the changes in winter to be rather continuous. The mechanism in spring is much clearer (more ozone in polar region => changes in SWHR gradient, etc) but largely depend on the winter circulation perturbation that is presently not easy to understand.

We agree with the reviewer that our explanation of the winter mechanism is more speculative than for the spring one; we have tried to stress that in our manuscript (see end of the last paragraph in Sect. 6.1 of the old manuscript version, which reads: "The details of this sensitivity are, however, difficult to diagnose using our experiments and this hypothesis should be subject to further examination") and we are sorry to hear we failed to convey this message more clearly. We have analysed the monthly evolution of specific quantities to examine the seasonal march of the signal, but identifying clearly and confidently the initial trigger is not easily possible as the monthly mean results are fairly consistent with each other. A more confident attribution of the initial triggering process responsible to the SH dynamical response in PHOT-ONLY would involve performing more specifically designed sensitivity simulations, which is beyond the scope of this manuscript. We do, however, think that our results at present constitute an important motivation for investigating the role of solar-induced ozone feedback in more detail, as it is a subject that has not been thoroughly acknowledged in previous literature.

We have now changed the manuscript, as to make this even clearer: we have toned down some of the statements about the mechanism responsible for the winter response, and stress that our suggestions/hypotheses should be followed up with further sensitivity experiments. Also, we include a discussion of an additional potential triggering process, i.e. the role of zonally-asymmetric ozone heating in modifying the wave-mean flow interactions. Evidence of the role of such ozone heating in modulating the NH polar vortex has been shown in the literature (e.g. Nathan and Cordero, 2007, Kuroda et al., 2007, 2008, McCormack et al., 2011, Silverman et al., 2018). It is plausible that the increased ozone levels in PHOT-ONLY have a similar effect in our study, with the zonally-asymmetric component of ozone heating being most important in early

winter (as opposed to the zonally-symmetric one in spring described in Sect. 6.2 of our manuscript due to increased ozone levels at high latitudes).

We have also investigated the interannual variability in the August zonal wind anomaly, and we include the histogram below to the Supplement and refer to it in the text. As shown in Fig. R1 below, the integrations suggest that it is both the mean behaviour and the extremes that shift, although longer model runs would be required to distinguish better differences in the distributions, especially in their tails.

[Figure]

Figure R1. Histograms of August monthly zonal mean zonal wind [ms$^{-1}$] at 1hPa and 49°S in the model experiments for (top) SMAX and (bottom) SMIN. The panels show (left) PHOT-ONLY, (middle) RAD-ONLY and (right) INTERO3, respectively.

3/ In light of my previous comment, I wonder if some of the identified changes between the PHOT-ONLY vs. RAD-ONLY & INTERO3 may not partly come from the fact that (if I understood correctly the experimental design) PHOT-ONLY MAX/MIN pair has a constant TSI-induced heating (since the radiation module solar forcing is fixed) while this is not the case for the two other experiments since between MAX & MIN conditions, the TSI-induced changes are considered in the radiation module. Could that lead to some bottom-up residual influence (even if the SSTs are fixed) and contribute to some of the identified differences? Is there a way to diagnose this? Do you think that this could have an influence?

This is an interesting suggestion. The fact that SSTs are prescribed and fixed in the experiments diminishes substantially the bottom-up response as only land temperatures can adjust. Hence,

the mechanisms for a bottom-up response to solar forcing which have been discussed in the literature will largely not be active here, e.g. a response in the tropical Pacific SSTs and links to the Walker and Hadley circulations. We note that the yearly mean SMAX-SMIN zonal mean temperature changes simulated in the troposphere are very small (Fig. 1). To remove entirely any bottom-up response would require us to fix land temperatures, which is very difficult to implement in the HadGEM3 model. Hence, we cannot rule out a potential role for a bottom-up influence, although the analysis of the experiments points to this being less important than the top-down influence from the stratospheric changes.

The radiation code takes TSI and partitions it into the shortwave radiation bands; hence it would be difficult in this model to keep TSI fixed whilst altering the distribution of solar energy across the UV part of the spectrum.

**Specific comments:**

P1, L21-24. As described in comment 2, I am not convinced yet that the SWHR gradients in winter play an important role.

We have changed the abstract in line with our response to the Reviewer's main comment 2 above.

P2, L26-28. Please give one or two example of "...specific aspects of model design" to make the issue more concrete.

We have included a couple of examples, i.e. the resolution of the radiation scheme and the height of the model top.

P3, L11-12: Indeed, the SH is less studied than NH, but there are clearly more studies than the one cited here (e.g. Petrick et al. (2012, JGR), see also numerous studies of Yuhji Kuroda and co-authors (the most recent by Kuroda was published this year in JGR)). Please cite some.

We have added a citation to Petrick et al., 2012; Kuroda et al. and Shibata, 2006, Kuroda et al., 2007; and Kuroda and Deuschi, 2016.

P3, L25: the "main" or the "only"?

This sentence now reads: "Unlike in Bednarz et al. (2016), however, the model version used here does not include the coupling of stratospheric aerosols with the radiation and photolysis schemes."

P3, L21-P4, L7. These 2 § are misleading since they leave the impression that transient simulations are performed. For instance, it is mentioned that HadISST are used and an 11-year forcing is implemented while it's not really the case. The authors should make clear from the beginning that they do idealized experiments that look at SMAX-SMIN conditions of the amplitude of the 11-year solar cycle. Presently, it is too confusing in my opinion.

As per the response to the Reviewer's main comment 1, we have now rephrased the text to clarify the scope of the model experiments and their design.

P4, L23-24 "The third pair, PHOT-ONLY SMAX/SMIN, is analogous to RAD-ONLY SMAX/SMIN, but the solar cycle forcing is included exclusively in the photolysis scheme while constant TSI and SSI are used in the radiation scheme." As mentioned in the main comment 3/, could the TSI change between SMAX and SMIN in the RAD-ONLY also be responsible for the observed difference in the signals? Would it not have been an option to keep always the TSI

constant? Note that this would have also made the use of the same fixed climatological SST prescribed for SMAX and SMIN more adequate in the case of the RAD-ONLY and INTERO3 experiments.

**See the response to the Reviewer's main comment 3. We also note that the use of fixed TSI in the radiation scheme is not straight forward to implement in our model at present as the change in partition of solar spectral irradiance over the shortwave radiation wavelength bins varies as a function of TSI.**

P6-7, section 3 and Figure 1. It would be relevant, I think, to calculate also the statistical significance of the difference between the RAD+PHOT & INTERO3 responses (panel d of Figure 1). This would strengthen the results and help motivating the analysis that are carried out later on in the paper.

**Considering the standard error associated with the RAD+PHOT response defined as a square root of the sum of squared standard errors associated with each RAD-ONLY and PHOT-ONLY responses, we find that the confidence intervals (±2 standard errors) around the RAD+PHOT and INTERO3 overlap. Therefore, the difference of these responses is not significant in a strict statistical sense. We now state that in the manuscript.**

**We note, however, that combining the errors associated with each of the RAD-ONLY and PHOT-ONLY responses by construction leads to broader confidence intervals than it is the case for each individual experiment pair alone, since each is affected by internal variability. Hence this is a more difficult criterion to pass.**

**Nevertheless, the yearly mean temperature difference between RAD+PHOT and INTERO3 in the SH high latitudes shown in Fig. 1d does largely exceed ±2 standard errors of the INTERO3 response. We have added this to the manuscript.**

P8, L20-24. That is very interesting to notice. Could that be due to the extraction method too used in the case of transient experiments and reanalysis and e.g. the difficulty to separate the solar signal from contributions of other variability factors? (see e.g. Chiodo et al., 2014, ACP)

**Indeed – this is partially what we refer to when noting possible contributions from interannual variability in that sentence.**

P9, L15-16. It appears from Figure S1 that Chapman dominates over NOx. This could be clarified in the text by giving the contribution of each e.g. in %.

**We have added this to the manuscript.**

P9, L27. The "small overestimation" is also statistically significant at the 2-sigma level near the peak at ~36 km. That means that even in the tropic, the RAD+PHOT contributions are not exactly linearly additive. This is I think still important to highlight and it shows that the complexity of the system needs to be accounted for, even in regions where we usually believe that the response is simple. Of course it's not major, but still worth mentioning.

**We have changed this sentence to: "There is some overestimation of the summed response compared with the control case; this illustrates that stratospheric ozone concentrations are controlled by a range of photochemical processes, thereby resulting in a complex dependence of the SMAX-SMIN ozone response on the associated temperatures, incoming wavelength-dependent solar radiation as well as any resulting changes in ozone columns above."**

P10, L19-23. What about the comparison with the results of Bednarz et al., 2018? Does the comparison between the timeslice and transient experiments help to get further understanding of the opened issues listed here? As you refer to this comparison previously in the manuscript (P8, L20-24), it may be worth looking at it again here.

**The SH dynamical response diagnosed in the ensemble of transient runs described in Bednarz et al., 2018 consists of a poleward shift of the SH polar vortex in austral winter and its weakening in spring (not shown). As this behaviour could result from the difficulty of separating the solar cycle response from the effect of other time-varying drivers, e.g. GHGs and/or ODSs, we refrain here from making a comparison between these idealised timeslice runs with all forcings except solar held fixed and the transient experiments with varying GHGs, ODSs, SSTs, sea-ice and stratospheric aerosols.**

P10, L26-30. It may be relevant to add the climatologies to the plots (as black contours on the background similarly to Fig. 1d). That would help to better visualize the jet strengthening and eddy driven jet displacement.

**We have now added the climatologies.**

P11, L4-7. Did you also look at the inter-annual variability? Is there a difference between the different runs? Are there SSWs–like perturbations (even though they should be rare in SH) that may be responsible for the SH easterly anomalies of the PHOT-Only experiment?

**See the response to the Reviewer's comment 2. We now include the histogram shown above to the Supplement, and we refer to it at the end of this paragraph.**

Figures 5 and 6. Similarly to Figure 1d, I think that the statistical significance of the differences could be relevant to show here.

**As it was the case with the yearly mean SH high latitude temperature response in Fig. 1d, the ±2 standard error confidence intervals around the RAD+PHOT and INTERO3 responses overlap. Thus, the difference of these is not significant in a strict statistical sense. We now state that in the manuscript. We note that the differences between RAD+PHOT and INETERO3 in Fig. 5 and 6 are nonetheless largely big enough to exceed the confidence interval (±2 standard errors) around the control INTERO3 response.**

P15, L19-20. Please indicate the altitudes of the peak (lower mesosphere, upper stratosphere is somehow vague). Where in the mesosphere does SWHR peak in RAD-Only (is 60 km the maximum?)?

**We have added this to the manuscript.**

P15,L24-P16,L2 & schematic on Fig. 8. As explained in comment 2/, that paragraph is not clear and too speculative to me. Please consider either making further analysis to support the present discussion or just tone down.

**Please see our response to the main comment 2.**

P16,L6. Instead of "primary driver", I would rather say that it's considered as the "initial driver".

**We prefer to stick to saying 'primary driver' as to indicate that this driver is usually considered as the main, if not the only, driver of the solar response during the whole dynamically active season.**

P16, L32-P17,L3. The arguments in this paragraph are not really convincing to me, since despite the fact that SWHR gradient are additive in JJA, the temperature and wind responses are not.

**As described in the manuscript, the non-additive nature of the temperature and wind responses must reflect contributions from dynamical processes which could be part of a non-linear response, as we discuss, and/or with some contribution from internal variability. We do point out that the magnitude of the non-additive component of the temperature and zonal wind response in JJA (Fig. 5) is relatively small here.**

**RESPONSE TO THE REVIEWER 2**

The manuscript attempts to disentangle the contributions from direct radiation and chemical effects of the solar irradiance. The authors applied the chemistry-climate model (CCM) UM-UKCA to simulate the steady-state atmospheric state for the solar maximum and minimum conditions. To separate the role of different processes and estimate the linearity of the overall response, the authors performed three pairs of experiments switching on only the direct influence of solar irradiance on radiation (RAD-ONLY), photolysis (PHOT-ONLY) and both (INTERO3). Several similar studies have been published before. The novelty of the manuscript consists of the application of more sophisticated model and the results concerning the non-linearity of the two considered processes during the southern hemisphere cold season. This conclusion is important for the community because it emphasizes the necessity of the interactive ozone treatment in the climate models. However, this important conclusion is also a weakest part of the manuscript because the identification of the responsible mechanisms is not convincing. This issue should be clarified before the publication of the manuscript, otherwise this important conclusion will not be fully appreciated.

**We thank the reviewer for the positive review and constructive comments that have improved the manuscript. Our replies to the individual comments are shown below in blue.**

**Major issues**

1. The chain of physical/chemical processes leading to the weaker polar vortex during SON is not convincingly presented. From the presented results, it is more or less clear that the story should start before the winter time. The gradient in heating rate for the PHOT-ONLY case should related to ozone gradient. It cannot be dramatically different from RAD-ONLY case, because the ozone increase inside polar vortex due to enhanced solar UV should not be large. During SON the obtained gradient in ozone is high, but I think it is rather related to dynamical processes during late winter. This dynamically induced increase of the ozone in the stratosphere produce strong heating rate gradients during SON and produce further suppression of polar night jet. Thus, the triggering process is not identified leading to weak understanding of the obtained results. I do not know which process can be involved, but I think the authors should try hard to find it.

**Please see our response to the Reviewer's 1 main comment 2.**

**Also, we note here that our hypothesis about the role of shortwave heating rate gradient in winter is based on the altitude of the maximum gradient change, which differs between PHOT-ONLY and RAD-ONLY due to heating rate differences in the tropics (See Fig. S1a of the old Supplement)**

2. The linearity of the atmospheric response to radiation and chemical processes was discussed in several previous publications. Maybe it is better to concentrate on discovered no-linearity in the southern hemisphere and makes the description of the annual and tropical mean responses shorter.

**We do try to concentrate on the SH dynamical response in our manuscript, however we think that some description of the annual and tropical mean response is useful here as (i) it shows that as far as the aspects of the stratospheric response to solar forcing already discussed in other studies our UM-UKCA response is not contrastingly different, (ii) we note a few points less frequently discussed in the context of the solar cycle before, e.g. the role of different chemical cycles for the solar-cycle induced ozone response, or the fact that while the SW heating rate response in PHOT-ONLY is higher that RAD-ONLY in the upper stratosphere, the corresponding temperature response there is lower (thereby illustrating the contribution of longwave heating rate change and any indirect dynamical processes in determining the tropical temperature response to the 11-year solar cycle). We have nonetheless attempted to shorten this section.**

**Minor issues:**

1. Figure 1: Statistical significance is missing on panel d).

**See our response to the same point raised by Reviewer 1 above.**

C22. Section 4.1: I recall the contribution of radiation and ozone effects were analyzed in Forster et al. (2011). Maybe it makes sense to mention this paper?

**Forster et al. (2011) is a stand-alone version of Chapter 3 of SPARC (2010), and we cite this study in the manuscript.**

3. Page 10, Section 5: I think it is pointless to carefully compare the results of time-slice model runs with permanent solar max/min conditions with observations and try to explain the difference.

**We agree but we do nonetheless make a brief comparison here in order to put our model results into context.**

4. Page 15: The explanations in the last paragraph are too vague and not instructive to my taste. These processes definitely exist, but it is not easy to illustrate how they work.

**Please see our response to the Reviewer's 1 main comment 2.**

5. Page 21, line 10: I wonder how it is possible keep this paper under review for already 3 years. It seems something is wrong with it. I would not site unpublished papers, because the results could be wrong.

**We have removed this reference.**

6. Page 23, line 3: I do not see clear time line of the changes. It looks like triggering mechanism is missing

**Please see our response to the Reviewer's 1 main comment 2.**

[revised manuscript text omitted]
).~~As a result, the SWHR response in the mid-stratosphere is significantly larger in PHOT-ONLY (e.g. by a factor of ~2 at 40 km).~~ Thus, while the contributions from the photolysis and radiation schemes to the SWHR changes are similar near the stratopause, the impact of the enhanced photochemical production of
20  ozone dominates in the mid-stratosphere (in agreement with Shibata and Kodera, 2005, and SPARC, 2010.

The tropical mean SWHR response in INTERO3 reaches up to ~0.16 K day$^{-1}$, and mostly follows the sum of PHOT-ONLY and RAD-ONLY (green line in Figure 2a). Thus, in the tropics, the individual SWHR responses in the single forcing
25  experiments can be added linearly to give an estimate very close to the full response.

**4.2. Temperature**

The corresponding SMAX-SMIN tropical average temperature responses  are shown in Fig. 2b (where $\Delta$TSI = 1.06 Wm$^{-2}$). In INTERO3, the maximum temperature response peaks at ~0.6 K over a fairly broad layer spanning ~45-60 km. Noteworthy, despite the identical implementation of the 11-year solar cycle forcing
30  in the model, the maximum response simulated in these timeslice runs is somewhat smaller than the response found in the analogous transient UM-UKCA integrations discussed in Bednarz et al. (2018, ~0.8 K/Wm$^{-2}$), likely indicating some contributions of indirect dynamical processes and/or interannual variability to one or both responses. In both cases, the UM-

UKCA simulated temperature response is somewhat smaller than found in some reanalyses (e.g. Mitchell et al., 2015b; Bednarz et al., 2018); this could be due to  large uncertainties in the responses diagnosed from reanalyses and/or some deficiencies in the model implementation of the solar cycle forcing (see Bednarz et al., 2018, for details).

Our integrations show  significant SMAX-SMIN changes in the upper stratospheric temperatures in RAD-ONLY and PHOT-ONLY,  illustrating that the solar cycle impacts on both atmospheric heating and photolysis are important in determining the temperature response there. As noted earlier, there is a large spread in the simulated upper stratospheric
10 temperature responses to the 11-year solar cycle forcing among different atmospheric models (e.g.: Austin et al., 2008; SPARC, 2010; Mitchell et al. 2015a; Hood et al., 2015). Thus, details of both  schemes in models and their implementation of the solar cycle forcing can have a strong influence on the simulated stratospheric temperature response to the 11-year solar cycle, and thus to  contribute to the inter-model spread.

15 The estimated standard errors in the magnitude of the temperature responses are comparatively larger than those found for the SWHRs, presumably owing to the additional contribution from dynamical processes to the stratospheric temperature variability through adiabatic heating/cooling. Thus, the temperature responses in RAD-ONLY and PHOT-ONLY are statistically indistinguishable throughout most of the stratosphere. We note that although PHOT-ONLY shows a somewhat stronger SWHR response in the upper stratosphere
20 than RAD-ONLY (Fig. 2a), the associated PHOT-ONLY temperature response there is smaller (Fig. 2b). This illustrates that the atmospheric temperature response to the amplitude of the 11-year solar cycle forcing is not only  controlled by changes in SWHRs, but also reflects the associated changes in the longwave component as well as any indirect changes in the circulation (not shown). As discussed above, the combined RAD+PHOT stratospheric temperature response in the tropics is in good agreement with the results from INTERO3 (consistent with
25 Shibata and Kodera, 2005 Gray et al., 2009, and Swartz et al., 2012).

**4.3. Ozone**

Figure 2c shows the simulated changes in the tropical mean ozone mixing ratios. In RAD-ONLY, we find a small SMAX-SMIN ozone decrease (up to ~0.5 %) in the mid-to-upper stratosphere and lower mesosphere . This results from the enhancement of chemical ozone loss under increased temperature, most importantly through the
30 Chapman and $NO_x$ ozone loss cycles (Fig. S1, Supplement, with the change in ozone loss via the Chapman cycle being a factor of ~1.5-6 larger between 40-50 km than via the $NO_x$ cycle; see also e.g., Barnett et al., 1975; Haigh and Pyle, 1982; Jonsson et al., 2004). In contrast, ozone increases in PHOT-ONLY throughout most of the stratosphere and lower mesosphere. This occurs primarily due to the enhanced photolysis of oxygen at wavelengths shorter

than ~242 nm (Eq. 1) and the subsequent formation of ozone (Eq. 2), but is also influenced by a solar-induced reduction in the stratospheric $NO_x$ levels (not shown), likely related to its enhanced photochemical removal (e.g. Soukhodolov et al., 2016). The  maximum tropical mean stratospheric ozone response in PHOT-ONLY (~3%) is somewhat larger than~~peaks in the mid-stratosphere with a magnitude of ~3 %. In comparison, the tropical ozone responsereaches up to (~~(~2.5 %), reflecting the inverse dependence of ozone on the associated temperature changes (with the temperature-induced modulation of the $NO_x$ cycle playing the dominant role in the mid-stratosphere Fig. S1, Supplementary Information, see also Jonsson et al., 2004). In the tropics, the yearly mean RAD+PHOT ozone response  is in a reasonable agreement with the response in INTERO3  (in agreement with  Swartz et al., 2012). The  some overestimation of the summed response compared with the control case this illustrates that stratospheric ozone concentrations are controlled by a range of photochemical processes, thereby resulting in a complex dependence of the SMAX-SMIN ozone response on the associated temperatures, incoming wavelength-dependent solar radiation as well as any resulting changes in ozone columns above.

To summarise, in the tropics the SMAX-SMIN changes in the SWHRs, temperature and ozone in  PHOT-ONLY and RAD-ONLY , which include the solar cycle forcing only in the photolysis and radiation schemes, respectively, can be summed linearly to give a response that is in a good agreement with the full response in the control INTERO3 pair.  Our UM-UKCA results  agree with the previous FDH calculations of Shibata and Kodera (2005)  Gray et al (2009) and SPARC (2010) as well as with the CCM results of Swartz et al. (2012). However, as noted above, the results show larger differences between the combined and the control temperature responses at high Southern latitudes (Fig. 1d). The following section analyses the corresponding responses modelled during the SH winter andspring, where the role of dynamical processes in modulating the response to solar cycle forcing has been shown to be important (Kuroda and Kodera, 2002; Kodera and Kuroda, 2002).

**5. The seasonal response in the Southern Hemisphere**

The mechanism proposed by Kuroda and Kodera (2002) and Kodera and Kuroda (2002) (thereafter referred to as KK2002a and KK2002b) to explain the dynamical response to the 11-year solar cycle forcing they identified in reanalysis data postulates that solar-induced changes in the tropical SWHRs and temperatures initiate a chain of feedbacks that modulates the strength of the polar vortex during the dynamically active season. The UM-UKCA simulated changes in zonal mean zonal wind and temperature during SH winter (June-August, JJA) and spring (September-November, SON) for the three pairs of  experiments are shown in Figs. 3 and 4, respectively.

The SMAX-SMIN differences in zonal mean zonal wind modelled in the SH high latitudes in INTERO3 are fairly weak and not highly statistically significant in either winter or spring (panels e-f in Figs. 3-4). There is a suggestion of a weak (~0.5 m s$^{-1}$) strengthening of the polar vortex near the stratopause during winter, consistent with the strengthened horizontal temperature gradient. In comparison, the reanalysis data suggest a strengthening of the SH polar jet on its equatorward side and weakening on its poleward side in winter; this spatial pattern is followed by an enhanced weakening/warming of the vortex in austral spring (e.g.: KK2002a; KK2002b; Frame and Gray, 2010; Mitchell et al., 2015b; Kodera et al., 2016). The disagreement between the model results and reanalysis data could be due to a number of factors, including: i) the uncertainties in the reanalysis SH response; ii) differences between the timeslice runs here with prescribed climatological SSTs/sea-ice and a transient evolution of the real atmosphere and its coupling to the oceans; iii) a positive bias in the model winter/springtime SH zonal wind climatology (not shown), which may affect interactions between planetary waves and the mean flow.

In RAD-ONLY, the zonal mean SH zonal winds in winter in the SH strengthen between SMAX and SMIN on the equatorward flank of the stratospheric/lower mesospheric jet by up to ~3 m s$^{-1}$ (Fig. 3a). This is associated with a cooling of the high latitude stratosphere by up to ~0.75 K (Fig. 4a). The strengthening of the polar vortex in the mid-latitudes extends down to the extratropical troposphere, where it is accompanied by a small (~0.5 m s$^{-1}$) negative zonal wind anomaly in the subtropical troposphere. The latter is indicative of a small poleward shift in the mid-latitude eddy-driven jet (Haigh et al., 2005; Simpson et al., 2009). Whilst the modelled stratospheric responses in RAD-ONLY are generally not highly statistically significant, they bear some resemblance to those found in reanalysis studies (e.g.: KK2002a; KK2002b; Frame and Gray, 2010; Hood et al., 2015; Mitchell et al., 2015b; Kodera et al., 2016). No significant high latitude response was simulated in RAD-ONLY in austral spring (panel b in Figs. 3-4).

In contrast, in PHOT-ONLY there is a strengthening of the stratospheric jet on its poleward side (up to ~1 m s$^{-1}$) and a weakening on its equatorward side (up to ~2.5 m s$^{-1}$) during SH winter (Figs. 3c and 4c). This represents a poleward contraction of the polar vortex, and is accompanied by a warming in the mid-to-upper high latitude stratosphere of up to ~1 K. Importantly, the easterly zonal wind anomaly develops with time, with significantly weaker zonal wind (up to ~3.5 m s$^{-1}$) simulated in the SH mid-to-high latitude upper stratosphere and lower mesosphere in spring (Fig. 3d). Coincident with the zonal wind changes, the winter Antarctic stratosphere is warmer by up to ~2 K in the austral spring (SON) mean (Fig. 4d). This modulation of the polar vortex persists until the vortex breaks up. A histogram showing the interannual variability of the mid-latitude zonal winds in August simulated in all runs is shown in Fig. S2, Supplement.

[Figure]

zonal wind change

a) RAD-ONLY, JJA    b) RAD-ONLY, SON

c) PHOT-ONLY, JJA    d) PHOT-ONLY, SON

e) INTERO3, JJA    f) INTERO3, SON

u [m/s]

[Figure]

**Figure 3.** Shading: sSeasonal mean (left: JJA and right: SON) SH zonal mean zonal wind change [m s⁻¹] between SMAX and SMIN for (a-b) RAD-ONLY, (c-d) PHOT-ONLY and (e-f) INTERO3. Single and double hatching indicates statistical significance

[Figure]

a) RAD−ONLY, JJA

b) RAD−ONLY, SON

c) PHOT−ONLY, JJA

d) PHOT−ONLY, SON

e) INTERO3, JJA

f) INTERO3, SON

[Figure]

**Figure 4. As in Fig. 3, but for the SMAX-SMIN zonal mean temperature changes [K] (shading) and climatological zonal mean temperature in SMIN run (contours). Contours spacing is 20 K (beginning at 140 K).**

The poleward shift of the stratospheric vortex simulated during winter in PHOT-ONLY and its equatorward strengthening in RAD-ONLY are essentially opposite to one another. Therefore, there is a substantial cancelation between the responses upon linear addition of the JJA means. The combined  RAD+PHOT temperature and zonal wind responses in JJA are generally similar to the weak response in INTERO3 (Fig. 5).

[Figure]

**Figure 5. (a) Shading shows the JJA mean difference between the sum of the RAD-ONLY and PHOT-ONLY temperature [K] responses and INTERO3. (b) as in (a) but for the corresponding zonal wind [ms⁻¹] responses. Contours in (a-b) show the responses in INTERO3 for reference.**

[Figure]

**Figure 6. As in Fig. 5 but for the SON mean.**

In austral spring (SON), a different picture emerges for the comparison between the sum of the PHOT-ONLY and RAD-ONLY responses and the INTERO3 response. In particular, the development of a significantly weaker and warmer polar vortex in PHOT-ONLY in spring contrasts strongly with the small circulation changes found in RAD-ONLY and INTERO3. Consequently, the combined RAD+PHOT responses in SON show larger differences compared to INTERO3 (Fig. 6). There is a difference in polar temperatures of up to ~1.75 K between the summed and INTERO3 responses, which is large enough to exceed the ±2 standard error confidence interval around the INTERO3 response and be evident in the annual mean (Fig. 1d). We note, however, that this SON difference between RAD+PHOT and INTERO3 is not significant in a strict statistical sense where the confidence intervals around both responses are considered. Nonetheless, our UM-UKCA results highlight that stratospheric high latitude dynamical responses to the amplitude of the 11-year solar cycle forcing are complex and could be non-additive. We explore this behaviour next.

**6. Proposed mechanism for the non-linear SH springtime response**

The mechanism for an 11-year solar cycle modulation of the polar vortex proposed by KK2002a/b centres on the direct solar-induced warming in the tropical region in autumn/early winter and the immediate changes in the horizontal temperature gradients as the primary driver of the chain of feedbacks between planetary waves and the mean circulation throughout the winter. To understand the potential reasons for the different dynamical responses simulated among our UM-UKCA experiments, we focus here on the changes in the SWHRs, the primary driver of the anomalous temperature tendencies. We use a simple measure for the solar-induced changes in the horizontal SWHR gradient across the SH as given by Eq. 4:

$$\Delta_{SMAX-SMIN}SWHR_{gradient} = \Delta_{SMAX-SMIN}SWHR_{0-60°S} - \Delta_{SMAX-SMIN}SWHR_{60°S-90°S} \quad (4)$$

~~First, we consider why the two single-forcing experiment pairs (RAD-ONLY and PHOT-ONLY) show contrasting SH winter polar vortex responses. We find that during winter (JJA), the maximum changes in the SWHR gradient near 60°S (Fig. 7a) in our runs peak at different altitudes, with the strongest changes in gradient found in the lower mesosphere in RAD-ONLY and in the upper stratosphere in PHOT-ONLY. Little insolation reaches the SH high latitudes in winter, and thus the SWHR responses there are small (Fig. S2c, Supplementary Information), so that the changes in the horizontal gradients in winter are dominated by the SWHR responses in the SH tropics/mid-latitudes (Fig. S2a). The latter are largely similar to those found for the tropical annual mean in Fig. 2a, following the same arguments as in Sect. 4.1.~~
~~We find that the development of the SH zonal wind and temperature anomalies in our experiment pairs is associated with changes in planetary wave propagation and breaking: the wave propagation/breaking is increased in PHOT-ONLY and reduced in RAD-ONLY, with no well-defined changes in INTERO3 (Fig. S3 and S4, Supplementary Information). To our knowledge few studies have examined the role of the spatial structure of the anomalous solar-induced tropical temperature tendencies for the resulting high latitude dynamical response (e.g. Ito et al., 2009, who looked at horizontal structure). We~~

suggest that the propagation and breaking of planetary waves within the stratosphere is sensitive to the spatial, in our case the vertical, structure of the anomalous SWHRs. These would act to alter temperature tendencies, thereby influencing zonal winds and potential vorticity gradients that are important for planetary wave propagation. The details of this sensitivity are, however, difficult to diagnose using our experiments and this hypothesis should be subject to further examination. The schematic representation of such mechanism is shown in Fig. 8a,c.

**6.12. SH spring**

[revised manuscript text omitted]

**ozone change**

a) RAD−ONLY, JJA

b) RAD−ONLY, SON

c) PHOT−ONLY, JJA

d) PHOT−ONLY, SON

e) INTERO3, JJA

f) INTERO3, SON

$O_3$ [%]

**Figure 9.** Seasonal mean (left: JJA, right: SON) SH  zonal mean changes in ozone mixing ratios [%] between SMAX and SMIN for (a-b) RAD-ONLY, (c-d) PHOT-ONLY and (e-f) INTERO3. Single and double hatching indicates statistical significance at the 90% and 95% confidence level, respectively (t-test) Note the additional contours at ±0.5 %.

[Figure]

**Figure 10. The monthly mean evolution of the polar cap average (90°S-60°S) change in the vertical component of the Transformed Eulerian mean circulation, $\overline{w}^*$ [mm s⁻¹], between SMAX and SMIN for (a) RAD-ONLY, (b) PHOT-ONLY and (c) INTERO3. Positive values indicate anomalous upwelling, and vice-versa. Thick white and grey lines indicate statistical significance at the 90% and 95% confidence level, respectively.**

**7. Discussion**

Haigh (2010) pointed out that the solar cycle induced ozone response alters the penetration of solar radiation to lower altitudes and, therefore, leads to a stratospheric SWHRs response that is complex and non-linear, depending on the associated changes in ozone. In agreement, our results indicate that the changes in ozone associated with photochemical production and coupling to the circulation,  not only in the tropics/mid-latitudes  but also in the polar regions, are

important for modulating the SH dynamical response to the amplitude of the 11-year solar cycle. A similar conclusion was reached by Hood et al. (2015) who suggested that both increases in tropical ozone as well as dynamically-induced sharp horizontal ozone gradients at higher latitudes are important for horizontal temperature gradients in the stratosphere and thus play a role in amplifying the associated seasonal zonal wind response. In addition, Kuroda and Kodera (2005), Kuroda and Shibata (2006) and Kuroda et al. (2008) found that under higher solar activity any changes in polar ozone driven by the Brewer-Dobson circulation during winter can persist in the lower stratosphere for several months, thereby inducing local temperature and zonal wind responses. In agreement, while the dynamical response simulated in RAD-ONLY largely disappears by spring, the response in PHOT-ONLY develops with time, with changes in polar ozone potentially contributing to this behaviour.

The importance of the high latitude ozone changes for the polar vortex strength and temperature was recently demonstrated by Karami et al. (2015) using simulations with prescribed idealised ozone anomalies, albeit not in the context of the 11-year solar cycle but relating to energetic particle precipitation events (see e.g. Gray et al., 2010). The role 
[revised manuscript text omitted]